**Machine Learning Assisted Chemical Characterization and Optical Properties of Atmospheric Brown Carbon in Nanjing, China**

Yu Huang[1], Xingru Li[2], Dan Dan Huang[3], Ruoyuan Lei[1], Binhuang Zhou[1], Yunjiang Zhang[1], Xinlei Ge[1, 4*]

[1] Jiangsu Key Laboratory of Atmospheric Environment Monitoring and Pollution Control, Collaborative Innovation Center of Atmospheric Environment and Equipment Technology, School of Environmental Science and Engineering, Nanjing University of Information Science and Technology, Nanjing 210044, China

[2] Analytical Instrumentation Center, Department of Chemistry, Capital Normal University, Beijing, 100048, China

[3] Shanghai Academy of Environmental Sciences, Shanghai 200233, China

[4] School of Environment and Energy Engineering, Anhui Jianzhu University, Hefei 230601, China

*Corresponding author: Xinlei Ge (Email: caxinra@163.com)

**Abstract**: The light-absorbing organics, namely brown carbon (BrC), can significantly affect atmospheric visibility and radiative forcing, yet their chemical and optical properties remain poorly understood. Here, a comprehensive analysis was conducted on the particulate matter ($PM_{2.5}$) samples collected in Nanjing, China during 2022 ~ 2023 with a particular interest on the identification of key BrC molecules. First, the water-soluble organic aerosol (WSOA) was more oxygenated during cold season (CS) due to a highly oxidized secondary OA (SOA) factor that was strongly associated with aqueous/heterogeneous reactions especially during nighttime, while the WSOA during summer season (SS) was less oxygenated and the SOA was mainly from photochemical reactions. Fossil fuel combustion hydrocarbon-like OA was the largest and dominant contributor to the light absorption during CS (55.6 ~ 63.7%). Secondly, our observations reveals that aqueous oxidation can lead to notable photo-enhancement during CS, while photochemical oxidation on the contrary caused photo-bleaching during SS; Both water-soluble and methanol-soluble organics had four key fluorophores, including three factors relevant with humic-like substances (HULIS) and one protein-like component. Thirdly, molecular characterization show that CHON compounds were overall the most abundant species, followed by CHO and CHN compounds, and significant presence of organosulfates in CS samples reaffirmed the importance of aqueous-phase formation. Finally, building upon the molecular characterization and light absorption measurement results, the machine learning approach was applied to identify the key BrC molecules, and 31 compounds including polycyclic aromatic hydrocarbons (PAHs), oxyheterocyclic PAHs, quinones, and nitrogen-containing species, etc., which can be a good reference for future studies.

## 1 Introduction

In ambient air, some organic aerosol (OA) species can absorb light in the near-ultraviolet (UV) and visible spectrum, and are termed as brown carbon (BrC) (Andreae and Gelencsér, 2006; Chen et al., 2020b). The BrC absorption exhibits strong wavelength-dependence that typically the absorption increases as the wavelength decreases (Laskin et al., 2015). A prior study has reported that BrC is responsible for approximately 40 % of UV-Vis light absorption (Yan et al., 2018), and thus BrC can play a crucial role in global climate and air quality (Jo et al., 2016; Feng et al., 2013). For examples, some studies show that the global radiative forcing of BrC ranges from approximately 0.22 to 0.57 W $m^{-2}$, equivalent to 27 ~ 70 % of that of black carbon (BC) (Lin et al., 2014; Zhang et al., 2017). Given such importance, recently many researches have been conducted to characterize the optical properties, sources, as well as chemical composition of BrC.

The sources of atmospheric BrC are highly complex, as it can originate from multiple primary emissions (Hecobian et al., 2010; Chakrabarty et al., 2010; Gu et al., 2022) as well as various secondary chemical processes (Wang et al., 2021) (Fleming et al., 2020; Jiang et al., 2021; Chen et al., 2020b). The primary sources mainly include coal combustion, biomass burning, and vehicular emissions (Wang et al., 2016; Sun et al., 2016; Qi et al., 2019; Chen et al., 2018; Gu et al., 2022); besides, a significant presence of chromophores originating from the ocean has been observed, indicating that the ocean/marine emission is likely also an important source of BrC (Cavalli et al., 2004). As said, secondary BrC species can be generated from many processes, for instances, the aromatic secondary OA (SOA) species formed under high $NO_x$ concentrations (Jaoui et al., 2006), reaction products of biogenic or anthropogenic SOAs with nitrogen-containing substances such as $NH_3$ and $NH_4^+$ (Updyke et al., 2012; Shapiro et al., 2009; Bones et al., 2010), and aqueous-phase reaction products from various carbonyl/phenolic precursors in cloud, fog, and aerosol water (Hu et al., 2017; Ye et al., 2018; Wang et al., 2021; Li et al., 2023; Ou et al., 2021). The light absorption properties of BrC are also closely related with its sources. Recent studies have linked

BrC light absorption with its various sources (both primary and secondary) deconvoluted from factor analysis of OA data determined by the aerosol mass spectrometry (AMS) (Chen et al., 2020b; Zhong et al., 2023; Chen et al., 2016), and provided the mass absorption efficiency (MAE) of individual BrC source/factor. In addition, the fluorescent properties are also investigated, which identified different types of humic-like substances (HULIS) and protein-like species as the key components (Xie et al., 2020; Chen et al., 2020a; Chen et al., 2021).

Essentially, the light absorption properties of BrC are governed by its chemical constitution. Current studies have identified several key classes of light-absorbing organics in atmospheric aerosols, such as the aromatic carboxylic acids, phenols, nitroaromatic compounds (NACs), polycyclic aromatic hydrocarbons (PAHs) and their derivatives (Lin et al., 2018; Huang et al., 2018; Wang et al., 2021; Xing et al., 2023; Gu et al., 2022; Chen et al., 2020b; Kuang et al., 2023). Some lignin pyrolysis/burning products including coumarins, flavonoids, stilbenes, and several sulfur-containing species are also found as significant BrC constituents (Fleming et al., 2020; Budisulistiorini et al., 2017; Huang et al., 2022). Xing et al. (2023) identified a series of BrC chromophores, encompassing nitrophenols, benzoic acids, oxygenated PAHs, phenols, aryl amides/amines, phenylpropene derivatives, coumarins and flavonoids, pyridines, and nitrobenzoic acids. Nevertheless, knowledge regarding the molecular composition of BrC so far is still incomplete and the aforementioned identified species only occupy a limited fraction of the BrC total light absorption. For examples, Zhang et al. (2013) measured eight NACs in Los Angeles and found that they contributed about 4 % of water-soluble BrC light absorption at 365 nm; Huang et al. (2018) measured 18 PAHs and their derivatives in Xi'an and found that they accounted for on average ~ 1.7 % of the overall absorption of methanol-soluble BrC; Gu et al. (2022) quantified eight NACs present in $PM_{2.5}$ samples collected during winter in Nanjing, which together could account for at most ~9 % of the total BrC absorption at 365 nm.

Emerging non-targeted approaches based on gas chromatography (GC) or liquid chromatography (LC) coupled with high-resolution mass spectrometry can detect

hundreds to thousands of molecules in OA (Kuang et al., 2023; Mao et al., 2022), enabling the identification of potential BrC species by connecting them with light absorption of OA. However, these approaches often output high-dimensional data with numerous variables, which must be evaluated appropriately. Traditional statistical methods often perform poorly when handling large datasets and fail to accurately identify complex relationships between variables (Fasola et al., 2020). Machine learning (ML) is a powerful tool that can effectively recognize nonlinear relationships between variables and address issues of collinearity among them (Tang et al., 2024). For instances, Zhang et al. (2023) employed the Random Forest (RF) algorithm to quantify the factors driving $PM_{2.5}$ trends in six cities on Tibetan Plateau from 2015 to 2022, revealing the importance of anthropogenic emission reductions; Wang et al. (2022a) integrated the positive matrix factorization (PMF) with a multi-layer perceptron (MLP) neural network to analyze the sources of BrC light absorption in six major Chinese cities, which finds that primary emissions, including biomass burning, vehicle emissions, and coal combustion, significantly contribute to BrC in these cities, while secondary processes contributed more significantly to light absorption in southern cities than in northern cities.

In this study, we conducted a systematic investigation on the chemical and optical properties on the fine particular matter ($PM_{2.5}$) samples in both daytime and nighttime collected in Nanjing, China during summer and cold seasons of 2022 ~ 2023. Particularly, for the first time, we applied the ML RF algorithm to connect the light-absorbing characteristics with the determined organic molecular identities, to assist the screen of key BrC molecules. Our findings regarding the BrC properties, and especially the BrC molecules proposed here can be a good reference for future studies.

## 2 Experimental methods

### 2.1 Sampling site and sample collection

The $PM_{2.5}$ filter samples were collected in the Nanjing, China, from July 11 to August 23, 2022, November 30 to December 10, 2022, February 13 to February 20,

2023, and March 3 to March 31, 2023. The first period represents the hot summer season (SS) (81 samples), and the later three periods represent the cold season (CS)(83 samples); note samples were not collected during precipitation events in both seasons. The sampling site was located inside the campus of Nanjing University of Information Science and Technology (32°12'20.82"N, 118°42'25.46"E). The site was in a suburban area, surrounded by residential buildings, and close to traffic arteries, and industrial zones (including chemical engineering and petrochemical plants, power plants and ironmaking and steelmaking plants).

A high volume sampler (Jinshida Ltd. Qingdao, China, model KB-1000) with a flow rate of 1.05 $m^3$ $min^{-1}$ was employed. $PM_{2.5}$ samples were collected on the prebaked (450 ºC) quartz fiber filters (Pallflex, USA, size of 8 × 10 inch). Daytime samples were collected from 08:00 to 18:00 (Local Beijing time), and nighttime samples were collected from 19:00 to 07:00 on the next day. Each filter was wrapped in an aluminum foil and kept frozen at -20 °C until analysis. The concentrations of common gas pollutants ($SO_2$, $NO_2$, CO, and $O_3$) were obtained from the nearby National Environmental Monitoring Center (http://www.cnemc.cn/), while the meteorological parameters (air temperature, relative humidity, wind speed and direction) were recorded in the same site as $PM_{2.5}$.

### 2.2 Chemical analyses

#### 2.2.1 Measurements of inorganic ions, organic carbon (OC) and elemental carbon (EC)

A number of round pieces (20 mm diameter) were punched from each sample filter, and were extracted by using 50 mL of ultrapure water (18.25 MΩ cm) (10 pieces) and methanol (4 pieces), respectively. The filter pieces underwent 30 minutes of sonication and were filtrated through the polytetrafluoroethylene (PTFE) syringe filters (0.22 μm) to remove insoluble materials. Cations ($NH_4^+$, $Na^+$, $K^+$, $Mg^{2+}$, $Ca^{2+}$) were measured using a 881 Compact IC pro ion chromatography (Metrohm, Switzerland), anions ($NO_3^-$, $SO_4^{2-}$, $Cl^-$, $F^-$) are determined by the ICS2100 (Dionex, USA). The water-

soluble organic carbon (WSOC) (μg C $m^{-3}$) was measured by a total organic carbon (TOC) analyzer (TOC-L, Shimazu, Japan). Operational details of these analyses can be found in our previous work (Chen et al., 2020a).

Concentrations of total elemental carbon (EC) and organic carbon (OC) in samples were measured using a thermal optical carbon analyzer (RT-4; Sunset Laboratory, USA) on a separate round filter piece (17 mm diameter) by using the IMPROVE TOT protocol (Bai et al., 2020). In addition, the residual OC and EC contents in samples after methanol extraction were determined with the same method mentioned above, and were subtracted from the total OC content, to derive the methanol-soluble OC (MSOC) (μg C $m^{-3}$).

### 2.2.2 Bulk analysis of organics

We employed specially an Aerodyne soot particle AMS (SP-AMS) to determine the bulk composition of water-soluble OA (WSOA) (Onasch et al., 2012). The analysis procedure is similar to that described in Ge et al. (2017). In brief, eight round pieces (20 mm diameter) of each filter were sonicated in 40 mL of ultrapure water, and the aqueous extract was nebulized using an atomizer (TSI, Model 3076), then the mist was dried by a silica gel diffusion dryer and the remaining particles were sent to the SP-AMS. The SP-AMS was operated in a laser-off mode, therefore to measure non-refractory species that can be rapidly vaporized at 600 °C (SP-AMS oven temperature). Note the SP-AMS employs a 70 eV electron impact (EI) ionization scheme, therefore the vaporized species are fragmented into positively charged ions with specific mass-to-charge (*m/z*) ratios and we obtained the composition of WSOA in the form of lumped molecular fragments rather than detailed molecular composition.

The SP-AMS data were post-processed using the Igor-based ToF-AMS analysis toolkit (SQUIRREL version 1.56D and PIKA version 1.15D), Elemental ratios including hydrogen-to-carbon (H/C), oxygen-to-carbon (O/C) and nitrogen-to-carbon (N/C) as well as the organic mass to organic carbon (OM/OC) ratio were calculated by using the methods proposed in Aiken et al. (2008), Canagaratna et al. (2015) and Ge et al. (2024). The WSOA mass concentration of each sample was normalized by

multiplying the WSOC concentration with its corresponding OM/OC. Then, we conducted the PMF analysis to resolve the sources of WSOA by utilizing the PMF evaluation toolkit (Version 2.06) (Ulbrich et al., 2009), followed strictly the protocol described in Zhang et al. (2011). As usual, we included only ions with $m/z \leq 120$, and PMF solutions were explored by varying the number of factors (from 3 to 8) and the rotation parameter ($f_{peak}$, from -1 to 1 with an increment of 0.2). Based on the diagnostic plots in Fig. S1 in the supplement, the four-factor solution was selected as the best solution. The four factors include a hydrocarbon-like OA (HOA) relevant with fossil fuel combustion, a biomass burning-related OA (BBOA), a less oxidized oxygenated OA (OOA1) and a more oxidized oxygenated OA (OOA2) (see details in Sect. 3.1.2).

### 2.2.3 Molecular characterization of organics

Molecular-level characterization of organic species was conducted by using an ultra-high performance liquid chromatography with a quadrupole time-of-flight (QTOF) mass spectrometer (UPLC-QTOF-MS) (ACQUITY UPLC H-Class coupled with a Xevo G2-XS QTOF, Waters). The sample pretreatment was described in Text S1. Compound separation was performed with a Luna Omega 1.6 μm C18 column (100 mm × 2.1 mm × 1.6 μm), and the sample aliquot was subjected to electrospray ionization (ESI), and detected in both positive and negative ion modes. The scanning *m/z* range for each mass spectrum was 50-1200, with a scanning rate of one spectrum per 0.1 second. More details are presented in Text S2.

The original UPLC-QTOF-MS data were processed using the Mass Spectrometry-Data Independent Analysis (MS-DIAL, version 4.92) software (Tsugawa et al., 2015), including peak extraction, alignment and deconvolution, achieving a detection probability of 70% in all samples for any identified compound. The method of systematic error removal using random forest (SERRF, a ML algorithm), was then introduced to reduce systematic errors and normalize the measured data (Fig. S2). All deconvoluted spectra were imported into the SIRIUS (Version 5.6.2) toolkit (Dührkop et al., 2019) to determine molecular formulas. The semi-quantitative concentrations of identified molecules were expressed in the normalized peak areas (NPRs), defined as

their peak areas acquired from SERRF divided by air volumes of the samples.

In addition, the double-bond equivalent (DBE) was used to indicate the level of unsaturation of the compound (Bae et al., 2011), and the aromaticity equivalent (Xc) (Yassine et al., 2014) was used to indicate the molecular structure. O/C, H/C and DBE values of the sample were averaged over all identified molecules based on their relative abundances (See details in Text S3).

### 2.3 Optical analyses

#### 2.3.1 Light absorption properties

The light absorption spectra of WSOC and MSOC in 200 ~ 800 nm were obtained using a UV-Vis spectrophotometer (UV-3600, Shimadzu, Japan) with a 0.5 nm interval. The absorbance at a certain wavelength λ ($A_\lambda$) were corrected by subtracting that at 700 nm ($A_{700}$) (near zero, as background), and the corresponding light absorption coefficient ($Abs_\lambda$, M $m^{-1}$) is calculated as below (Hecobian et al., 2010):

$$Abs_\lambda = (A_\lambda - A_{700}) \times \frac{V_l}{V_a \times L} \times \ln(10) \tag{1}$$

Where $V_l$ represents volume of the extract (water or methanol), $V_a$ denotes air volume of the filter piece, and L is the optical path length (0.01 meters here).

The corresponding mass absorption efficiency ($MAE_\lambda$, $m^2$ $g^{-1}$) can then be calculated below:

$$MAE_\lambda = \frac{Abs_\lambda}{[WSOC]\ \ ([MSOC])} \tag{2}$$

Where [WSOC] ([MSOC]) represents the mass concentration of WSOC (MSOC). Following previous studies (Laskin et al., 2015; Chen et al., 2018; Xie et al., 2020; Chen et al., 2020b), the light absorption at 365 nm ($Abs_{365}$) was employed as a surrogate for BrC in this work.

The relationship between the Absorption Ångström Exponent (AAE)(an index of the wavelength dependence) (Andreae and Gelencsér, 2006) and light absorption is shown below:

$$Abs_\lambda = K \cdot \lambda^{-AAE} \tag{3}$$

Where *K* is a constant related to light absorption, and we computed the AAE values

in the 300 ~ 450 nm range.

The direct radiative forcing effect of BrC can be represented by the simple forcing efficiency (SFE) (in W $g^{-1}$), which is the energy added to the earth-atmosphere system per unit mass of aerosol (Bond and Bergstrom, 2006). The SFE of BrC at the wavelength λ can be expressed below (Chen and Bond, 2010):

$$\frac{\mathrm{dSFE}}{\mathrm{d}\lambda} = -\frac{1}{4}\frac{\mathrm{dS}(\lambda)}{\mathrm{d}\lambda}\tau_{\mathrm{atm}}^2(\lambda)(1-F_{\mathrm{C}}) \\ [2\times(1-\alpha_{\mathrm{S}})^2\beta(\lambda)\mathrm{MSE}(\lambda) - 4\times\alpha_{\mathrm{S}}\times\mathrm{MAE}(\lambda)]\mathrm{d}\lambda \quad (4)$$

Where S(λ) represents the solar irradiance at λ obtained from the ASTM G173–03 reference spectra. $\tau_{atm}$ denotes the atmospheric transmission (0.79), $F_C$ is set to 0.6, indicating the fraction of cloud cover, the global average value of $\alpha_s$ is fixed at 0.19, representing the surface albedo, $\beta$ is the backscatter fraction, and MSE and MAE are the mass scattering efficiency and mass absorption efficiency of BrC, respectively. .

When estimating the radiative effect of BrC, the direct radiative forcing caused by aerosol scattering can be neglected. Therefore, the absorbed radiative forcing within a given spectral range is calculated by the simplified Eq. (5):

$$\mathrm{SFE} = \int \frac{\mathrm{dS}(\lambda)}{\mathrm{d}_\lambda}\tau_{\mathrm{atm}}^2(\lambda)(1-\mathrm{F}_C)\times\alpha_{\mathrm{S}}\times\mathrm{MAE}(\lambda)\,\mathrm{d}\lambda \quad (5)$$

### 2.3.2 Fluorescence properties

Characterization of excitation-emission matrix (EEM) of the extracts was performed using a fluorescence spectrophotometer (Cary Eclipse, Agilent, USA) The wavelength range of excitation was set from 230 to 500 nm, and that of emission was from 250 to 600 nm, the scanning resolutions of excitation and emission were 5 nm and 2 nm, respectively, with the scanning speed of 1200 nm $min^{-1}$. The photomultiplier tube (PMT) detector voltage was set at 600 V. The measurement was subjected to instrument calibration, internal filter correction, Raman/Rayleigh scattering correction, and all EEM spectra were subjected to blank filter subtraction. The processed data were further analyzed using the parallel factor analysis (PARAFAC) to group potential components with similar fluorescent properties. The analysis was performed using MATLAB 2022b software with the drEEM toolbox (Murphy et al., 2013).

Fluorescent properties of the extracts were also characterized by the humification

index (HIX), biological index (BIX), and fluorescence index (FI). HIX is defined as the ratio of integrated fluorescence emission intensity in the range of 435 - 480 nm to that in the range of 300 to 345 nm when excited at 254 nm; BIX is calculated as the ratio of emission intensity at 380 nm to that of 430 nm for the excitation wavelength of 310 nm; FI is the ratio of emission intensity at 470 nm to that of 520 nm under a fixed 370 nm excitation wavelength (Birdwell and Engel, 2010; Mcknight et al., 2001).

**2.4 Machine learning screening of key light-absorbing species**

The ML RF model was used here to screen the key light absorbing species by linking the target variable ($Abs_{365}$) with the identified organic molecules (in NPRs), via the "randomforest()" function in R software (Version 4.3.2). The model included 500 decision trees and estimated the variance through a cross-validation during training. The dataset was divided into a training set (80% of total) and a test set (20% of total) to assure accuracy and robustness of the model.

The model outputted two key indices to assess the importance of each molecule to the light absorption. One metric is IncNPu_val, which can measure the purity of nodes. During the construction of each tree in the RF model, each split can increase the purity of nodes, therefore if more frequently a variable is used in splitting, more contribution it has to the increase of purity of nodes, then the variable is considered to be important. IncMSE_val is another index based on the mean squared error (MSE). When we permute a variable, increase in the projected error can serve as a measure of its importance. If a variable with a significant impact on the predicted results is permuted, the model's MSE would increase significantly, resulting in a high IncMSE_val value (González et al., 2015). Under the $50^{th}$ percentile of IncMSE_val, some variables had zero or even negative contributions to IncMSE_val. Considering the definition of IncMSE_val, such variables would have either no or negative influence on model fitting, thus only the top 50 % of compounds were chosen. Afterwards, intersection of the two indices were considered as potential BrC chromophores.

Moreover, a molecule typically requires a substantial uninterrupted conjugation on its molecular backbone to effectively absorb visible light (Lin et al., 2018), therefore a

compound with the ratio of DBE to carbon (DBE/C) greater than that of linear polyenes (with a molecular formula of $C_xH_{x+2}$, DBE/C = 0.5) (Cain et al., 2014) is treated as a potential BrC compound. Besides, the DBE/C ratio should be less than the upper limit of DBE for natural compounds (DBE/C = 0.9) (Lobodin et al., 2012). Finally, the candidate compounds passed aforementioned procedures were compared with those in open-source databases, including MoNA (https://mona.fiehnlab.ucdavis.edu/) and MassBankEU (https://massbank.eu/), to be interpreted as the key BrC compounds.

## 3 Results and discussion

### 3.1 Chemical properties

#### 3.1.1 General characteristics

During the sampling period, the temperatures were 34.34 ± 3.23 °C (daytime) ( ± one standard deviation, hereinafter) and 30.85 ± 2.97 °C (nighttime) during SS, 13.00 ± 6.04 °C (daytime) and 9.97 ± 5.04 °C (nighttime) during CS, and the relative humidity (RH) levels were 53.61 ± 10.33 % (daytime) and 65.88 ± 9.87 % (nighttime) during SS, 47.81 ± 18.94 % (daytime) and 55.56 ± 15.74 % (nighttime) during CS, respectively (Figs. 1a and b). Figure 1c depicts the temporal variations of different components. Average concentrations of OC, EC, WSOC, MSOC and total ionic species during daytime and nighttime in the two seasons are summarized in Table 1. Clearly, concentrations of all components were higher in CS than those in SS, but the daytime/nighttime differences were relatively small in both seasons. Also, the MSOC levels were larger than WSOC in all samples. MSOC occupied 82.4 % and 81.5 % while WSOC occupied 61.3 % and 49.5 % of the total OC during SS and CS, respectively, indicating that methanol can more effectively extract the aerosol organics than water.

The mean mass contributions of different ions to their total during SS and CS, respectively are shown in Fig. 1c too. The particles were overall neutral as the molar ratios of inorganic anions to cations were 0.97 and 0.98 during SS and CS, respectively (Fig. S3). The most abundant ion was sulfate in SS (45.5 %) and nitrate in CS (50.7 %), as low temperatures during CS favor the partitioning of nitrate to particle phase. As so,

the sulfur oxidation ratio (SOR, $[SO_4^{2-}]/([SO_4^{2-}]+[SO_2])$) and nitrogen oxidation ratio (NOR, $[NO_3^-]/([NO_3^-]+[NO_2])$) were 0.58 and 0.16 (daytime) and 0.56, 0.17 (nighttime) during SS, 0.40, 0.28 (daytime) and 0.42, 0.30 (nighttime) during CS, respectively. NOR was indeed much higher in CS especially during nighttime than those in SS. Furthermore, ammonium ($NH_4^+$) was the predominant cation while sulfate, nitrate and chloride were major anions. The scatter plots of molar concentrations of ammonium versus summed sulfate, nitrate and chloride (Fig. S4) reveal different bonding forms of the aerosol inorganic salts in different seasons. The correlations were both tight (correlation coefficients close to 1) yet the fitted slopes during SS were 0.80 (daytime) and 0.89 (nighttime) while those during CS were 0.98 (daytime) and 0.99 (nighttime), respectively. Such results demonstrate that ammonium was deficit to neutralize the cations therefore significant amounts of metal salts (such as sodium/calcium sulfate/nitrate) could exist during SS while during CS, most inorganic species were in the forms of $(NH_4)_2SO_4$, $NH_4NO_3$ and $NH_4Cl$ with no appreciable metal salts.

### 3.1.2 Features and sources of water-soluble organics

Regarding the water-soluble portion of organic species (WSOA), Figure 2a presents the average high resolution mass spectra (HRMS) during SS and CS, respectively. It can be seen that, WSOA during CS appeared to be much more oxygenated than that during SS (O/C: 0.58, 0.59 vs. 0.44, 0.45) . To further unravel causes of such differences, PMF analysis were conducted and the HRMS of resolved factors are presented in Fig. 2b, while mass contributions of these factors during SS and CS as well as the Pearson's correlation coefficents ($r$) of these factors with other components are illustrated in Fig. 3.

The HOA MS was dominated by $C_xH_y^+$ ions (57.2 %), such as $C_4H_7^+$ (*m/z* 55) and $C_4H_9^+$ (*m/z* 57), primarily originating from hydrocarbons emitted from fossil fuel combustion (such as traffic) (Canagaratna et al., 2004). Among the four factors, HOA exhibited the lowest O/C ratio (0.24) and the highest H/C ratio (1.66). The second factor was identified as BBOA, since it has distinct peaks at *m/z* 60 (mainly $C_2H_4O_2^+$, 0.76 % of the total MS) and *m/z* 73 (mainly $C_3H_5O_2^+$, 1.09 % of the total MS), which are

characteristic fragments associated with levoglucosan, a tracer compound of biomass burning particles (Kumar et al., 2022; Qin et al., 2017). Correlations between BBOA and these two tracer ions were indeed tight (0.71 with $C_2H_4O_2^+$and 0.82 with $C_3H_5O_2^+$). A notably positive correlation between BBOA and $K^+$ (Fig. 3b) further supports its BB origin as $K^+$ is also a common BB emission tracer (Yu et al., 2018). Note the BBOA here had a relatively higher O/C ratio of 0.61 than those identified in previous offline AMS measurements, such as Yangzhou (0.45) (Ge et al., 2017), Beijing, China (0.59) (Qiu et al., 2019), and Marseille, France (0.54) (Bozzetti et al., 2017), suggesting the potential presence of partially aged BBOA components in this factor.

The other two factors are secondary. OOA1 was less oxidized with a O/C of 0.39 and OOA2 was more oxygenated with the highest O/C of 0.65 among all factors. OOA1 had characteristic fragments at *m/z* 29 ($CHO^+$) and *m/z* 43 (mainly $C_2H_3O^+$) while OOA2 had the least fraction of oxygen-free $C_xH_y^+$ ions (28.5 %) but the largest fraction of oxygenated ions (32.8 % of $C_xH_yO_1^+$ and 18.9 % of $C_xH_yO_2^+$) among the four factors. OOA2 also correlated well with $CO_2^+$ (*m/z* 44) ion (*r* of 0.81), a characterisitic ion of highly oxygenated carboxylic/dicarboxylic acids. Moreover, OOA2 had the highest N/C of 0.095 as well as those of $C_xH_yN^+$ (9.1 %) (such as $CH_2N^+$, $CHN^+$ and $CH_4N^+$) and $C_xH_yO_zN^+$ (3.8 %) (such as $CHON^+$, $CH_2NO^+$ and $CH_4NO^+$) ions, indicating the presence of amines and amino acids correspondingly (Ge et al., 2024). Besides, the N/C level of OOA2 is close to that of fogwater observed in Fresno, indicating the aqueous phase reactions are likely an source of those nitrogen-containing ions in OOA2 (Kim et al., 2019). In addition, sulfur-containing organic ions (such as $CH_2S^+$, $CH_3SO_2^+$ and $CHS^+$ ) were almost exclusively present in OOA2 and in a significant fraction (2.7 %), as such ions were strongly associated with aqueous/heterogenous reactions (Zorn et al., 2008; Huang et al., 2020; Petters et al., 2021; Mcneill, 2015), reassuring that OOA2 was probably linked with aqueous/heterogenous formation pathway.

The time series of mass contributions of the four PMF factors are shown in Fig. 1d, and significant differences can be observed during the two sampling seasons, as can be seen clearly in Fig. 3a. HOA was a significant source in both SS and CS, and as

expected, it was higher during daytime due to the stronger traffic activities. BBOA was much less important than HOA, but its contribution during CS was obviously more than that during SS (12.5 ~ 13.0 % vs. 7.2 ~ 7.8 %), and accordingly, HOA contribution was slightly larger during SS than during CS (38.0 ~ 41.7 % vs. 29.9 ~ 36.8 %). The most striking difference lies in two SOA factors. OOA1 occupied nearly half of the total WSOA (47.5 ~ 49.8 %) while contributions of OOA2 were only 3.0 ~ 5.0 % during SS; on the other hand, OOA2 occupied 38.6 ~ 43.5 % of WSOA mass while those of OOA1 were down to 11.6 ~ 14.1 %. The much larger OOA2 fraction during CS explains its overall high oxidation degree depicted in Fig. 2. These results are well consistent with our previous studies, as we show that during summer in Nanjing, photochemical reactions dominate the SOA formation and yield relatively less oxygenated OA (Xian et al., 2023; Wang et al., 2022b), while during cold seasons, aqueous formation of SOA becomes more important which can generate highly oxygenated OA (Wu et al., 2021). The air temperature (a solar radiation indicator) and ozone (a photochemical product) both correlated positively with OOA1 but negatively with OOA2 (Fig. 3b), further verifying the dominance of photochemical pathway of OOA1 not OOA2. As is well known, particulate nitrate was strongly associated with heterogenous reactions and gas-to-particle partitioning favored by low temperature and high RH especailly during CS, and indeed, OOA2 correlated much tighter with $NH_4NO_3$ than OOA1 did.

## 3.2 Optical properties

### 3.2.1 Light-absorbing properties of WSOC and MSOC

The average light absorption coefficients of BrC in WSOC and MSOC in 300 ~ 700 nm during daytime and nighttime of SS and CS are illustrated in Fig. 4a. As expected, the values all exponentially decreased as a function of wavelength. The calculated AAE values are listed in Table 1. It is interesting to find that the WSOC AAE had no significant difference between SS and CS (6.35 vs. 6.43), while that of MSOC AAE appeared to be notable (5.99 vs. 6.89); Compared with the MSOC AAE, the WSOC AAE was higher during SS but smaller during CS. The AAE values obtained

here are slightly smaller than those reported in Beijing (7.3 ~ 7.5 of WSOC) (Du et al., 2014; Chen et al., 2016), comparable to that in Guangzhou (6.7 of WSOC) (Fan et al., 2016). Note Chen and Bond (2010) reports that particles generated from smoldering of various types of wood exhibit a AAE of 6.9 ~ 11.6 (MSOC); Lambe et al.(2013) shows that lab-generated secondary BrC possesses a AAE of 5.2 ~ 8.8 (MSOC). Compared with these values, our measured AAE values (<7) here probably suggest a dominance of secondarily formed BrC for both WSOC and MSOC. This can be verified by Fig. 3a for WSOC, as indeed it was dominated by SOA particularly during nighttime in both seasons. However, the large difference in WSOA chemical composition in different seasons (especially SOA proportions) did not result in a large difference in WSOC AAE, demonstrating clearly the non-correspondence of chemical species to light-absorbing species (a.k.a., BrC). On the contrary, for MSOC, the relatively large difference of MSOC AAE in different seasons likely reflect the distinction of BrC species but not necessarily chemical constitution. The light absorption coefficients at 365 nm ($Abs_{365}$) are listed in Table 1 too. The average $Abs_{365,\ WSOC}$ during CS (4.87 M $m^{-1}$) was approximately 2.15 times that of SS (2.27 M $m^{-1}$), and that of MSOC during CS (4.97 M $m^{-1}$) was also much larger than that during SS (3.64 M $m^{-1}$); Nighttime values were higher than those in daytime expect for WSOC during CS. For the same set of samples, $Abs_{365,\ MSOC}$ values were typically larger than $Abs_{365,\ WSOC}$ except that CS daytime $Abs_{365,\ MSOC}$ was slightly smaller than $Abs_{365,\ WSOC}$ (4.65 vs. 4.89 M $m^{-1}$). Scatter plots of $Abs_{365}$ versus WSOC (and MSOC), and $Abs_{365,\ WSOC}$ versus $Abs_{365,\ MSOC}$ for the four series of samples are given in Fig. S5. The correlations were generally well especially those of WSOC and MSOC ($r$ > 0.80), suggesting that there is a large overlap of extracted species between WSOC and MSOC, as well as their BrC constituents.

Regarding the MAE at 365 nm ($MAE_{365}$), $MAE_{365,\ WSOC}$ during CS (0.75 $m^2\ g^{-1}$) was higher than that during SS (0.55 $m^2\ g^{-1}$), indicating its stronger light absorption ability during CS; however, the $MAE_{365,\ MSOC}$, unlike $Abs_{365,\ MSOC}$, was smaller during CS than that during SS (0.50 vs. 0.72 $m^2\ g^{-1}$). Besides, $MAE_{365}$ values for both WSOC and MSOC were slightly larger during nighttime than those during daytime in both

seasons. Compared to previous winter studies, the $MAE_{365, WSOC}$ in Nanjing here was lower than that in Beijing (1.21 ~ 1.26 $m^2$ $g^{-1}$) (Du et al., 2014; Chen et al., 2016; Li et al., 2020), similar to our earlier observation in Yangzhou (0.75 $m^2$ $g^{-1}$), but the $MAE_{365, MSOC}$ appears to be less than that in Yangzhou (1.12 $m^2$ $g^{-1}$)(Chen et al., 2020). To further explain the low $MAE_{365}$ observed here, we investigated the air mass origins of our samples collected in different periods via back trajectory analysis (at an altitude of 200 m and 24 hours backwards) using the MeteInfo (Version: 3.0.0) (Wang, 2019). As shown in Fig. S6, only a limited fraction of air mass trajectories passed through sea and coastal areas (clusters 4 and 5, 27.05 % during daytime and 29.54 % during nighttime) during SS, while during CS, proportions of trajectories that intercepted sea/costal air increased to 79.80 % (clusters 1, 2 and 3 during daytime) and 69.44% (clusters 2, 3 and 4 during nighttime), respectively. Note the air masses during CS are somewhat unusual as typically they mainly originate from inland regions (Wu et al., 2019b), which might cause the low $MAE_{365}$ observed in this work as particles affected by marine air can be less light-absorptive than those influenced by inland air (Li et al., 2022).

Saleh (2020) proposes a method that uses the $MAE_{405}$ (MAE at 405 nm) – AAE two-dimension space to assess the light-absorbing ability of BrC, as shown in Fig. 4b. The majority of samples in this study fall into the regime of W-BrC (weakly light-absorptive BrC) with a few MSOC samples locating in the VW-BrC (very weak BrC) regime, which are similar to a few other observations (Zhou et al., 2021; Xu et al., 2022). The BrC in MSOC seemed to cover a broader region than it in WSOC, indicating that the MSOC BrC might contain a wider array of species and/or originate from more diverse sources/processes. Daytime/nighttime difference of MSOC BrC was also more obvious than that of WSOC BrC.

At last, we estimated the SFE values of WSOC and MSOC in the range of 300–700 nm, considering the actual visible light wavelength as well as the negligible light absorption above 700 nm of BrC. As summarized in Table 1, the mean $SFE_{MSOC}$ (2.43 W $g^{-1}$) during SS was higher than that of WSOC (2.20 W $g^{-1}$), but it became smaller during CS (2.23 W $g^{-1}$) and was much lower than that of WSOC (3.24 W $g^{-1}$). The

$SFE_{WSOC}$ values in both SS and CS were lower than that in Beijing (4.6 ± 1.7 W $g^{-1}$ in summer and 6.2 ± 2.0 W $g^{-1}$ in winter), especially in CS (Deng et al., 2022). For both WSOC and MSOC, SFE values during nighttime were slightly larger.

### 3.2.2 Source apportionment of light absorption of WSOC

In Sect. 3.1.2, sources of WSOA were identified and quantified, herein we applied a multiple linear regression (MLR) algorithm to apportion the light absorption of WSOC to these sources. The scatter plot of reconstructed $Abs_{365, WSOC}$ versus measured values are shown in Fig. S7. The fitted slope is 1.06 with a Pearson's *r* of 0.90, verifying the robustness of this method on our dataset. The calculated regression coefficients, representing the factors' $MAE_{365}$ values ($m^2$ $g^{-1}$) are listed in Table 2. Compared to our earlier results in Yangzhou (Chen et al., 2020b), the HOA $MAE_{365}$ (0.71 $m^2$ $g^{-1}$), was much less than that in Yangzhou (1.46 $m^2$ $g^{-1}$), while the BBOA $MAE_{365}$ values were similar (0.71 vs. 0.77); $MAE_{365}$ values of OOA1 (0.12 $m^2$ $g^{-1}$) and OOA2 (0.83 $m^2$ $g^{-1}$) were very close to the two SOA factors in Yangzhou (0.11 and 0.85 $m^2$ $g^{-1}$). However, here the less oxygented OOA1 has the small $MAE_{365}$ while in Yangzhou, the more oxygenated SOA has the similar $MAE_{365}$ as OOA1, and *vice versa* for the other pair. This work finds that the more oxidized SOA has a stronger light absorption ability, opposite to that reported in Yangzhou. Nevertheless, the two findings are not contradictory with each other, as atmospheric ageing can lead to either photo-enhancement or photo-bleachment, dependent upon the precursors. For instances, aqueous oxidation of BBOA can increase (Gilardoni et al., 2016) yet conversely aqueous processing of fossil fuel combustion OA can decrease the light absorptivity of OA (Wang et al., 2021). As discussed in Sect. 3.2.1, the unusual air masses during CS in this work clearly indicate different precursors from those in Yangzhou.

The average contributions of HOA, BBOA, OOA1 and OOA2 to $Abs_{365, WSOC}$ across the whole campaign were 33.05 %, 15.49 %, 6.00 % and 45.46 %, respectively (Table 2). Figure 5 further presents contributions of the factors under different scenarios. Compared with their mass contributions shown in Fig. 3a, during SS, the dominant contributor of light absorption became HOA (daytime 63.7 %, nighttime 55.6 %), and

contributions of BBOA and OOA2 both increased relative to their mass fractions; while OOA1's contribution was largely reduced to 15.0 ~ 16.2 % due to its small $MAE_{365}$. Previous studies have consistently identified coal combustion (Fan et al., 2016; Li et al., 2019; Song et al., 2019) and traffic emissions (Hecobian et al., 2010) as significant contributors to BrC, together with our results here, highlighting the substantial impact of anthropogenic fossil fuel combustion on atmospheric visibility. During CS, OOA2 dominated the light absorption (daytime 50.7 %, nighttme 63.0 %) owing to its large mass contribution as well as large $MAE_{365}$; OOA1 became a very minor contributor (2.1 ~ 2.8 %), HOA contribution decreased while BBOA contribution increased relative to their mass fractions. Overall, we find that primary fossil fuel combustion emissions govern water-soluble BrC light absorption during SS especailly during daytime, while during CS, secondary highly aged species (likely from aqueous/heterogenous reactions) dominates, especially during nighttime.

To further explore the impact of atmospheric ageing on BrC, we plotted $MAE_{365}$ as a function of O/C in Fig. 6. Interestingly, during SS, $MAE_{365}$ generally decreased with the increase of O/C, especially in daytime as its fitted slope of -1.56 was over 2 times that of nighttime (-0.76). On the other hand, $MAE_{365}$ showed an increasing trend against O/C during CS, particularly in nighttime as the fitted slope was 1.43, larger than that of daytime (1.12). These results further supports our earlier findings and underscore that during summer photochemical reactions can lead to photo-bleachment of aerosols while during cold seasons aqueous/heterogenous reactions might dominate the secondary formation and lead to photo-enhancement; clearly, photochemical oxidation and aqueous/heterogenous reactions are more active during daytime and nighttime, respectively, consistent with the slopes in Fig. 6. Also, photochemcially produced SOA was often less or moderately oxygenated and that from aqueous/heteregenous oxidation was more oxidized, and there is a turning point at O/C of 0.45 ~ 0.5 in the $MAE_{365}$-O/C plot, which was found in previous studies too (Zhong et al., 2023; Jiang et al., 2022).

#### 3.2.3 Fluorescent properties of WSOC and MSOC

The fluorescence indices like HIX, BIX and FI, can infer the types and sources of

dissolved organic matter (DOM) in aquatic systems and soils (Lee et al., 2013; Huguet et al., 2009). Recently, these indices have been employed to investigate sources and aging processes of atmospheric OA (Fu et al., 2015; Qin et al., 2018; Deng et al., 2022; Murphy et al., 2013). Here, we calculated these indices for both WSOC and MSOC.

HIX represents the degree of humification, and a high HIX means high aggregation, C/H ratio and aromaticity of the organics (Zsolnay et al., 1999; Mcknight et al., 2001; Birdwell and Engel, 2010), thus it normally increases upon ageing (Fan et al., 2019; Murphy et al., 2013). In this study, HIX of WSOC during SS and CS were on average 3.34 and 4.68, respectively (Table 3), much less than the HIX levels in aquatic or soil DOM (Dong et al., 2017), suggesting an overall low aromaticity of atmospheric OA in Nanjing. As a comparison, the WSOC HIX are higher than those in Colorado, USA (2.42) (Xie et al., 2016) and Tianjin, China (2.73 and 2.22) (Deng et al., 2022), but significantly lower than that in Nanjing during 2017-2018 (7.07) (Xie et al., 2020). An earlier study proposes the HIX ranges of 1.4-5 for fresh SOA and 4.2-6.1 for aged SOA (Lee et al., 2013). Despite influences of other primary sources, the average HIX during SS did fall in the fresh SOA range and the value during CS entered the edge of aged SOA, in line with the oxidation degrees of OA (Fig. 2a) and mass proportions of fresh/aged SOA factors (namely, OOA1/OOA2) (Fig. 3a) during different seasons. The average HIX of MSOC (2.72 and 3.48) were lower than those of WSOC in both seasons, indicating that HULIS with high aromaticity are preferentially soluble in water.

FI is indicative of the relative contributions of terrestrial and biogenic sources while BIX, in contrast to HIX, can be treated as a freshness index. A fluorophore is often associated with high aromaticity if FI is low (Fu et al., 2015), and a high BIX indicates a high content of freshly released organics (such as biological or microbial derived species) (Wen et al., 2021; Huguet et al., 2009; Murphy et al., 2013). The average WSOC BIX values during SS and CS were determined to be 0.84 and 0.88, respectively, with corresponding FI of 1.91 and 1.90 (Table 3), and the corresponding MSOC BIX were 0.90 and 0.96, and MSOC FI were 2.27 and 2.11, respectively. Compared with results during 2017-2018 in Nanjing, BIX and FI values of WSOC were

similar, yet those of MSOC here were larger (Xie et al., 2020). Figure 7 shows the measured data in the HIX-FI and HIX-BIX diagrams along with results from a few other studies. It can be seen that, almost all BIX values distributed in the range of 0.6 ~ 1 (Huguet et al., 2009) and FI values distributed within 1.6 ~ 1.9 (Mcknight et al., 2001), suggesting that OA in both seasons was influenced by a mix of terrestrial and microbial/biogenic sources. For both WSOC and MSOC, BIX was slightly higher during CS than during SS, attributing to the fact that OA during CS contained more aged SOA species. Nearly no difference for WSOC FI during different seasons were observed, but the MSOC FI during CS was slightly lower than that during SS, meaning that MSOC during CS had a high aromaticity as expected. In addition, BIX and FI values during nighttime were marginally higher than those during daytime in all cases.

#### 3.2.4 Identification of key fluorophores of WSOC and MSOC

The 3D EEM-PARAFAC analysis was adopted to identify the key fluorophores of BrC, with results in Fig. 8 and Fig. S8. Four components were resolved for both WSOC and MSOC. For WSOC, C1 exhibited a peak at Ex = 230 nm and Em = 374 nm, identified as less oxidized HULIS typically associated with combustion sources. Its contribution was only 4.9 % during SS but increased to 19.2 % during CS (Fig. S8a). C2 had a prominent peak at Ex = 230 nm and Em = 396 nm and a second peak at Ex = 320 nm and Em = 396 nm, classified as a HULIS-related component too, as the dual-peak distribution of fluorescence spectrum is often associated with HULIS (Coble, 1996; Murphy et al., 2011; Yu et al., 2015); its second peak indicates the abundance of compounds with condensed aromatics, conjugated bonds, and non-linear rings (Matos et al., 2015). C2 contribution was comparable during different seasons (37.5 % vs. 38.6 %), and it seemed to be more important in nighttime than in daytime (44.0 % vs. 26.9 % during SS, and 41.4 % vs. 34.8 % during CS). C3 component, with a peak at Ex = 240 nm and Em = 446 nm, was considered as a highly oxidized HULIs component, relevant with secondary processes (Cheng et al., 2016; Cao et al., 2021). Hawkins et al. (2016) and Aiona et al. (2017) reported fluorescent pattern of products from aqueous-phase reaction of aldehydes with ammonium sulfate or amines (Ex < 250/300 nm and

Em > 400 nm) well matches the pattern identified here. As discussed earlier, aqueous/heterogenous reactions contributed to WSOA, especailly during CS; correspondingly, C3 contribution was indeed much higher during CS than during SS (24.1 % vs. 14.7 %). C4, with a prominent peak at Ex = 230 nm and Em = 308 nm and a second peak at Ex = 275 nm and Em = 305 nm, was characterized as a protein-like component (Yan and Kim, 2017; Wu et al., 2019a; Chen et al., 2020b). C4 was the single largest contributor (42.9 %) during SS particualy during daytime (55.0 %), but became the least one during CS (18.0 %), indicating distinct fluorescent properties of OA in different seasons. Overall, since C1 ~ C3 are all relevant with HULIS, the WSOC fluorescent properties were governed by HULIS (57.1 % in SS and 82.0 % in CS).

Similarly for MSOC, three HULIS-related fluorophores (C1 ~ C3) and one proteinaceous fluorophore (C4) were separated (Fig. 8b). The spectral signatures between the two series of fluorophores were slightly different, with the MSOC peak excitation and emission wavelengths being a bit larger than those of WSOC, especailly for C2 and C3. Figure S8b shows contributions of the different components to MSOC fluorescene. C1 was much more important in MSOC (26.6 % in SS and 39.2 % in CS) than in WSOC, and became the largest contributor of MSOC during CS; summed C2 and C3 contributions (30.7 % in SS and 37.0 % in CS) were on the other hand much less than in WSOC; C4 remained to be the largest (42.6 %) similar to that in WSOC during SS. It is worth to point out that C4 not only contains proteinaceous species like tyrosine and tryptophan but also certain PAHs or phenolic substances emitted from fossil fuel and/or biomass burning, especially in urban aerosols (Barsotti et al., 2016; Chen et al., 2021; Chen et al., 2020b; Cao et al., 2021; Deng et al., 2022). Probably, the proteinaceous species dominated the fluorescence during SS for both WSOC and MSOC, while during CS PAHs and phenolic compounds became more important and they might prefer to dissolve in methanol therefore lead to a higher contribution in MSOC than in WSOC (23.8 % vs. 18.0 %). Daytime/nighttime variations of MSOC were similar to those of WSOC, as shown in Fig. S8.

### 3.3 Molecular composition of organics

#### 3.3.1 Overview of identified molecules

We classified the identified molecular formulas of organics via UPLC-QTOF-MS analysis into 8 categories, namely CH, CHO, CHN, CHS, CHON, CHOS, CHNS, and CHONS. Overall, the negative ($ESI^-$) and positive ($ESI^+$) ion modes identified 466 ~ 865 and 644 ~ 1065 formulas, respectively (details in Table S1). Figures S9 and S10 shows the number and signal fractions (relative abundances of signal intensities) of different classes of compounds, respectively.

Under $ESI^+$ mode, CHON compounds were the most abundant species in term of the number fraction – nearly half during SS (daytime 50.5 % and nighttime 46.9 %), and over half during CS (daytime 54.1 % and nighttime 55.1 %), and the abundance of its signal was even more prevailing – over half in all cases and up to 67.7 % during SS daytime; the second abundant species were CHO compounds, occupying 23.0 ~ 30.6 % of the total number of molecules and 15.4 ~ 21.9 % of the total signal intensity; CHONS and CHN species were the other two relatively abundant classes – together occupying ~ 20 % (number fraction) and ~ 10 ~ 24 % (signal fraction) of total identified compounds; contributions of other four classes of compounds were very minor, in terms of both number and signal intensity. Relatively, under $ESI^-$ mode, CHO compounds marginally prevailed over CHON compounds in number (36.2 ~ 44.4 % vs. 32.6 ~ 38.0 %), but during CS their signal fractions were still lower (33.3 ~ 35.1 % vs. 39.3 ~ 46.5 %). More enrichment of CHO compounds in $ESI^-$ than in $ESI^+$ is consistent with a previous work (Lin et al., 2012) as these compounds most likely contain carboxyl groups and are easily deprotonated in $ESI^-$ mode. Number fractions of CHONS compounds in $ESI^-$ mode were ~5 ~ 10% more than those in $ESI^+$ mode, while the most contrasting difference was that CHN compounds were rarely detected in $ESI^+$ mode, and instead CHOS compounds that were negligible in $ESI^-$ mode could be effectively detected in $ESI^+$ mode (3.9 ~ 4.8 % in number) and their signal fractions were more significant (6.4 ~ 17.4 %).

#### 3.3.2 CHO compounds

For the detected CHO compounds, we plotted them in the van Krevelen (VK) diagrams according to their H/C and O/C ratios in Fig. 9. Most molecules had the O/C ratios < 0.5, but a broad distribution of H/C ratios (0.5 ~ 2.0). Molecules with high H/C ratios (≥ 1.5) and low O/C ratios (≤ 0.5) (Region A) were typically associated with aliphatic compounds, while those with low H/C ratios (≤ 1.0) and low O/C ratios (≤ 0.5) (Region B) are usually assigned to oxygenated aromatics (Kourtchev et al., 2014). We further calculated the Xc values of all CHO compounds to investigate their molecular structures in Fig. 9. Clearly, saturated aliphatic CHO compounds (Xc < 2.5) were most abundant (323 out of 418 in $ESI^+$ mode, and 315 out of 481 in $ESI^-$ mode) and mainly distributed in Region A. Appreciate number of unsaturated compounds particularly those with benzene ring or naphthalene ring structures (2.5 <Xc <2.8), distributed across from H/C of 0.5 to 1.75 but with O/C < 0.5 in $ESI^-$ mode and from H/C of 0.1 to 1.75 but with a few in the side of O/C > 0.5.

The oxygenation state ($OS_c$) (Kroll et al., 2011) (defined as 2*O/C - H/C), is another metric to assess the ageing/oxidation degree of a compound. Figure 10 illustrates the dependence of $OS_c$ on carbon number for all CHO compounds. The molecules had a broad coverage of $OS_c$ (-2 to +2) and carbon number (up to 50). Kroll et al. (2011) grouped the compounds into different origins according to their $OS_c$ and C numbers, including fossil fuel combustion HOA, BBOA, semi-volatile oxygenated OA (SV-OOA, typically less-oxidized) and low-volatility oxygenated OA (LV-OOA, typically more-oxidized), as marked in Fig. 10. Obviously, for both $ESI^-$ and $ESI^+$ modes, a large portion of compounds belonged to HOA and BBOA. In $ESI^+$ mode, a significant portion of molecules located in the BBOA region, while in $ESI^-$ mode, more molecules tended to be found in the HOA region, and even more molecules located within HOA regime during CS than during SS (Figs. S11 and S12), indicating large influences from anthropogenic emissions. Besides, the number of nighttime LV-OOA molecules was more than that of daytime particular during CS, acting as a supporting evidence of aqueous/heterogeneous reactions.

### 3.3.3 CHON and CHN compounds

We mapped the detected CHON compounds colored by Xc in the VK diagrams shown in Fig. 11. The compounds were sorted into different series according to the functional groups as well. For ESI$^+$ mode (Fig. 11a), the compounds containing a -NO moiety were dominant and a majority of CHON compounds were saturated with Xc < 2.5. Among them, $C_6H_{15}NO(CH_2)_n$ might be N,N-diethylethanolamine homologous compounds, and $C_6H_{15}NO_2(CH_2)_n$ might be diisopropanolamine homologous compounds, as both compounds possess lone pair electrons, prone to positive charge (Ge et al., 2011). Unsaturated CHON compounds with $X_C \geq 2.5$ located in the bottom-left corner, such as $C_5H_5NO(CH_2)_n$, $C_7H_7NO(CH_2)_n$, $C_8H_7NO(CH_2)_n$, and $C_9H_7NO(CH_2)_n$, likely homologous compounds of hydroxypyridine, benzamide, 4hydroxy-benzene acetonitrile, and hydroxyquinoline, respectively (Ma and Hays, 2008; Wang et al., 2020). In ESI$^-$ mode, the compounds scattered wider than those in ESI$^+$ mode in the VK plot (Fig. 11b), and the majority of them contained one or two nitrogen atoms. Over 25 % of the CHON formulas can be classified as monocyclic or polycyclic compounds with Xc ≥ 2.5 (even up to 68 % during SS daytime; inferred from Figs. S13 and S14). The identified series of homologous compounds mostly situated in the bottom-left corner and also with Xc ≥ 2.5, such as $C_6H_5NO_3(CH_2)_n$, $C_6H_5NO_4(CH_2)_n$, $C_8H_7NO_3(CH_2)_n$, $C_8H_7NO_4(CH_2)_n$, and $C_{10}H_7NO_3(CH_2)_n$, likely nitrophenol, nitrocatechol, nitroacetophenone, nitrophenylacetic acid, and nitronaphthol homologues, respectively (Wang et al., 2018b; Song et al., 2019; Lin et al., 2017; Lin et al., 2015).

As stated in Sect. 3.3.1, CHN compounds were only enriched in ESI$^+$ mode. The scatter plot of H/C versus N/C of these compounds is depicted in Fig. 12 (results of different periods are shown in Fig. S15). Similarly, they are colored by Xc and sorted into a number of different series. Most of these compounds were amines with one or two N atoms. The series of aliphatic amines and other monocyclic species with $X_C$ < 2.71 mostly located in upper part of the plot, including $C_6H_{15}N(CH_2)_n$, $C_5H_{11}N(CH_2)_n$ and $C_6H_{12}N(CH_2)_n$, $C_4H_6N_2(CH_2)_n$, $C_5H_6N_2(CH_2)_n$, $C_7H_6N_2(CH_2)_n$, and $C_{11}H_{17}N(CH_2)_n$.

Note the presence of 2 N-heterocyclic species was a sign of presence of BBOA (Wang et al., 2017). The series of $C_{10}H_9N(CH_2)_n$ (1N-PAHs) with $X_C \geq 2.71$ may represent the aminonaphthalene homologues (Ge et al., 2011), likely from initial burning of carbonaceous materials (Mao et al., 2022).

#### 3.3.4 CHOS and CHONS compounds

Among the CHOS formulas (only significant in $ESI^-$ mode), ones with O/S ratios $\geq 4$ were classified as organosulfates (OSs), which were the most abundant type (Table 4). Its number fractions were particularly high during CS (daytime 54.3 %, nighttime 68.6 %), reiterating the importance of aqueous SOA formation during CS. For the CHONS compounds, in $ESI^-$ mode, 5.3 ~ 12.5 % of the formulas had O/(4S+3N) ratios $\geq 1$, allowing them to be assigned to $-OSO_3H$ and $-ONO_2$ groups, namely nitrooxy-organosulfates (nitrooxy-OSs) (Wang et al., 2018a); while in $ESI^+$ mode, 9.8 ~ 11.8% of total CHONS formulas were apportioned as nitrooxy-OSs (Table 4).

### 3.4 Machine learning assisted identification of key BrC molecules

As stated in Sect. 2.4, the ML RF algorithm was used to identify the key BrC chromophores, and we finally confirmed 31 compounds (18 in $ESI^+$ mode and 13 in $ESI^-$ mode); details regarding their molecular formulas and proposed structures, etc., are summarized in Table S2. These species are relevant with 4 out of 8 identified types of compounds (CH, CHO, CHN and CHON) (Fig. 13). Note except 6 out of the 31 species (4-methylcoumarin, urocanate, 3-hydroxybenzoic acid, chrysin, 2-hydroxypyridine and 4-hydroxyacetophenone), all other species were in general reported as BrC molecules before (See Table S2).

Two PAHs (acenapthylene and fluoranthene, belonging to CH category) in $ESI^+$ mode were identified, which is reasonable as PAHs are known important BrC (Aurell et al., 2015; Kuang et al., 2021).

Twelve CHO compounds (5 in $ESI^+$ mode and 7 in $ESI^-$ mode) were identified. In $ESI^+$ mode, 9-fluorenone and benzanthrone belonging to oxyheterocyclic PAHs (O-PAHs), are known as important BrC chromophores (Kuang et al., 2023); scopoletin is

also known as a light-absorbing compound (Zhang, 2018); phthalic anhydride is an oxygen-containing heterocyclic compound. A previous study reports that methanol (the solvent used here) might react with conjugated carbonyl species (such as phthalic anhydride, maleic anhydride, and maleimide) (Chen et al., 2022), thereby affecting the light absorption of relevant BrC species, further studies are needed to verify phthalic anhydride as a key chromophore. In ESI$^-$ mode, a pair of quinone isomers (1-hydroxyanthraquinone and 2-hydroxyanthraquinone) were resolved, in agreement with Kuang et al. (2023), which identified 1-hydroxyanthraquinone as a BrC chromophore in Beijing; 1-hydroxypyrene is a hydroxylated PAHs , also proven as a BrC before (Huang et al., 2022).

The identified seven CHN compounds (exclusively in ESI$^+$ mode) included 4 N-heterocyclics, 2 nitro-PAHs, and 1 quinoline compound. It is well known that biomass burning (BB) release a lot of BrC species. As mentioned earlier, CHN compounds are abundant in BB emissions (such as agricultural waste burning and forest fires (Laskin et al., 2009)); small N-containing heterocyclic compounds with one or two aromatic rings, can be effectively produced from thermal decomposition of plants (Ma and Hays, 2008), and high temperature pyrolysis of CHN compounds and N-containing plant materials, can result in N-PAHs (Lin et al., 2016). Therefore, identification of the CHN species here as key BrC chromophores are well justified.

The remaining confirmed key BrC molecules included 10 CHON compounds (4 in ESI$^+$ mode and 6 in ESI$^-$ mode). For ESI$^-$ mode, 3-hydroxyanthranilic acid is an amino phenolic compound $^-$, and the rest five compounds are all nitrophenols, well known as BrC (Li et al., 2020). Another amino phenolic compound, 2-aminophenol was identified in ESI$^+$ mode. Previously, efficient light absorption at 275 nm of 2-aminophenol has been reported, which can be further enhanced in the presence of $Fe^{3+}$ due to formation of oligomers (Al-Abadleh et al., 2022). For acridone in ESI$^+$ mode, earlier studies have shown that acridine exhibits increased light absorbance in the wavelength range of 260 ~ 320 nm under irradiation in $N_2$, air, or $O_2$; additionally, a deep yellow layer forms on the surface, indicating the production of light-absorbing

products, which was identified as acridone (Negron-Encarnacion and Arce, 2007).

## 4 Conclusions

This work performed a comprehensive investigation on the chemical and optical properties of BrC in ambient $PM_{2.5}$ samples. Regarding the chemical properties, it was found that methanol was able to extract more OC than water (~82% vs. 49.5 ~ 61.3% of total OC). The WSOA was composed of two primary factors relevant with fossil fuel combustion (HOA) and biomass burning (BBOA), and two SOA factors (a less oxidized OOA1 and a highly oxygenated OOA2). During CS, OOA2 was abundant (38.6 ~ 43.5 %) while during SS OOA1 was abundant (47.5 ~ 49.8 %); HOA was also an important contributor in both seasons (29.9 ~ 41.7%) but BBOA contribution was relatively minor (7.2 ~ 13.0 %). Further analyses reveal that OOA1 was mainly associated with photochemical reactions while OOA2 was strongly linked with aqueous/heterogeneous reactions. Regarding the light absorption, our observation shows that $Abs_{365, MSOC}$ was typically larger than $Abs_{365, WSOC}$, but though $MAE_{365, MSOC}$ was still larger than $MAE_{365, WSOC}$ during SS, it became smaller than the $MAE_{365, WSOC}$ during CS, likely owing to that the air mass trajectories during CS significantly intercepted sea/coastal air. The light absorbing abilities of both WSOC and MSOC were weak, but our observations suggest that aqueous oxidation can lead to significant photo-enhancement, therefore the light absorption of WSOA was dominated by OOA2 (50.7 ~ 63.0 %) during CS; while photochemical oxidation could cause a photo-bleaching effect and therefore the contribution of OOA1 to WSOA absorbance was small (15.0 ~ 16.2 %), and HOA contribution was prevailing during SS (55.6 ~ 63.7 %). PARAFAC analysis on the fluorescent spectra of WSOC and MSOC both resolved four key components with slightly differences, including three HULIS component and one protein-like component. HIX, BIX and FI indices also suggest that both WSOC and MSOC originated from a mix of terrestrial and microbial/biogenic sources.

The molecular analysis determined 644 ~ 1065 molecules in $ESI^+$ mode and 466 ~ 865 molecules in $ESI^-$ mode. Overall, CHON compounds were the most abundant type

especially in $ESI^+$ mode, while CHO compounds slightly exceeded CHON compounds in number but were still lower in signal intensity. CHN compounds was the third important class and only detectable in $ESI^+$ mode. The VK diagrams further demonstrate the different aromaticity equivalent (Xc) values and evolution pathways of the different classes of compounds. In addition, significant presence of organosulfates and nitroxy-organosulfates in CS samples especially during nighttime re-affirm the importance of aqueous-phase oxidation during CS. At last, based on the molecular characterization and light absorption measurement results, we applied the ML RF algorithm to identify the key BrC molecules, and we successfully identified 31 key species, including mainly the PAHs, oxyheterocyclic PAHs (O-PAHs), quinones and N-containing compounds. Overall, our findings presented here expand the scientific understanding regarding the chemical composition (both bulk and molecular level) and optical properties (both light absorption and fluorescence) of BrC, and are valuable to evaluate the impact on air quality and radiation balance of BrC. Besides, our identified list of key BrC molecules can be a useful reference for future studies.

**Code availability.** The software code to analyze the SP-AMS data is publicly available at: https://cires1.colorado.edu/jimenez-group/ToFAMSResources/ToFSoftware/index.html. The software code to analyze the UPLC-QTOF-MS data is publicly available at: https://systemsomicslab.github.io/compms/msdial/main.html. The software code using SERRF to normalize UPLC-QTOF-MS data is available at: https://slfan.shinyapps.io/ShinySERRF/

**Data availability.** The data in this study are available from the authors upon request (caxinra@163.com).

**Supplement.** The supplement related to this article is available online at: XXX

**Author contributions.** YH, XL and DDH conducted the experiments. YH, XL, RL, BZ, YZ and XG performed the data analysis. YH and XG wrote the paper. All authors reviewed the paper and provide useful suggestions.

**Competing interests.** The contact author has declared that neither they nor their co-authors have any competing interests.

**Disclaimer.** Publisher's note: Copernicus Publications remains neutral with regard to jurisdictional claims in published maps and institutional affiliations.

**Acknowledgements.** We sincerely thank the logistic help from the Center for Experimental Atmospheric Science and Environmental Meteorology of Nanjing University of Information Science and Technology (NUIST) during sampling.

**Financial support.** This work has been supported by the National Natural Science Foundation of China (grant nos. 42021004 and 22361162668).

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

Table 1. The average mass concentrations of major chemical components as well as the parameters of optical properties of $PM_{2.5}$ collected in Nanjing during two seasons.

| | Summer Season (SS) | | | Cold Season (CS) | | |
|---|---|---|---|---|---|---|
| | Daytime | Nighttime | Average | Daytime | Nighttime | Average |
| OC (μg $m^{-3}$) | 7.02±3.04 | 6.87±2.51 | 6.94±2.76 | 12.9±5.77 | 12.74±5.29 | 12.82±5.51 |
| EC (μg $m^{-3}$) | 1.13±0.31 | 1.22±0.37 | 1.17±0.34 | 1.94±0.93 | 1.6±0.74 | 1.77±0.86 |
| $AAE_{WSOC}$ | 6.34±0.65 | 6.35±0.69 | 6.35±0.67 | 6.43±0.68 | 6.44±0.86 | 6.43±0.77 |
| $AAE_{MSOC}$ | 6.02±0.90 | 5.96±0.96 | 5.99±0.92 | 7.06±0.94 | 6.70±0.65 | 6.89±0.82 |
| WSOC (μg $m^{-3}$) | 4.06±1.31 | 4.45±1.44 | 4.26±1.38 | 5.81±2.29 | 5.75±2.25 | 6.34±2.27 |
| MSOC (μg $m^{-3}$) | 5.64±2.12 | 5.79±1.89 | 5.72±1.99 | 10.42±4.98 | 10.49±4.57 | 10.45±4.75 |
| Total ions (μg $m^{-3}$) | 18.00±5.49 | 18.69±8.05 | 18.49±6.91 | 35.41±15.02 | 43.12±17.94 | 39.22±16.88 |
| $Abs_{365, WSOC}$ (M $m^{-1}$) | 2.15±0.90 | 2.38±0.80 | 2.27±0.85 | 4.89±2.63 | 4.86±2.46 | 4.87±2.53 |
| $Abs_{365, MSOC}$ (M $m^{-1}$) | 3.44±1.40 | 3.82±1.55 | 3.64±1.48 | 4.65±2.24 | 5.31±2.71 | 4.97±2.49 |
| $MAE_{365, WSOC}$ ($m^2$ $g^{-1}$) | 0.54±0.16 | 0.56±0.15 | 0.55±0.16 | 0.73±0.20 | 0.77±0.21 | 0.75±0.21 |
| $MAE_{365, MSOC}$ ($m^2$ $g^{-1}$) | 0.68±0.32 | 0.75±0.30 | 0.72±0.31 | 0.48±0.18 | 0.52±0.16 | 0.50±0.17 |
| $SFE_{WSOC}$(W $g^{-1}$) | 2.16±1.29 | 2.24±1.36 | 2.20±1.33 | 3.16±1.8 | 3.42±1.25 | 3.24±1.84 |
| $SFE_{MSOC}$(W $g^{-1}$) | 2.28±2.37 | 2.55±1.85 | 2.43±2.10 | 2.19±1.01 | 2.26±0.81 | 2.23±0.91 |

Table 2. Multi-linear regression results of the four factors and corresponding average contributions to the total light absorption of water-soluble organics (WSOA).

| Factor | Coefficients ($m^2 \cdot g^{-1}$) | | Contributions (%) |
|---|---|---|---|
| | Average | Standard error | |
| HOA | 0.71 | 0.11 | 33.05 |
| BBOA | 0.71 | 0.06 | 15.49 |
| OOA1 | 0.12 | 0.07 | 6.00 |
| OOA2 | 0.83 | 0.14 | 45.46 |

Table 3. The average values of fluorescence indices of both water-soluble organic carbon (WSOC) and methanol-soluble organic carbon (MSOC).

| | Summer Season (SS) | | | Cold Season (CS) | | |
|---|---|---|---|---|---|---|
| | Day | Night | Average | Day | Night | Average |
| $HIX_{WSOC}$ | 3.16±0.74 | 3.51±1.12 | 3.34±0.97 | 4.68±0.94 | 4.67±0.91 | 4.68±0.92 |
| $HIX_{MSOC}$ | 2.65±0.99 | 2.78±0.86 | 2.72±0.92 | 3.42±0.73 | 3.53±0.58 | 3.48±0.66 |
| $FI_{WSOC}$ | 1.85±0.16 | 1.97±0.17 | 1.91±0.18 | 1.89±0.11 | 1.92±0.09 | 1.90±0.10 |
| $FI_{MSOC}$ | 2.25±0.26 | 2.30±0.31 | 2.27±0.28 | 2.10±0.15 | 2.12±0.14 | 2.11±0.15 |
| $BIX_{WSOC}$ | 0.81±0.16 | 0.86±0.14 | 0.84±0.15 | 0.86±0.10 | 0.91±0.08 | 0.88±0.09 |
| $BIX_{MSOC}$ | 0.89±0.19 | 0.9±0.13 | 0.90±0.16 | 0.95±0.10 | 0.97±0.11 | 0.96±0.10 |

Table 4. The number percentages of organosulfates (OSs) in CHOS compounds in $ESI^-$ mode and those of nitrooxy-OSs in CHONS compounds in both modes.

| | SS | | CS | |
|---|---|---|---|---|
| | Daytime | Nighttime | Daytime | Nighttime |
| OSs ($ESI^-$) | 34.8% | 42.3% | 54.3% | 68.6% |
| Nitrooxy-OSs ($ESI^-$) | 5.3% | 12.0% | 11.0% | 12.5% |
| Nitrooxy-OSs ($ESI^+$) | 10.3% | 11.8% | 10.5% | 9.8% |

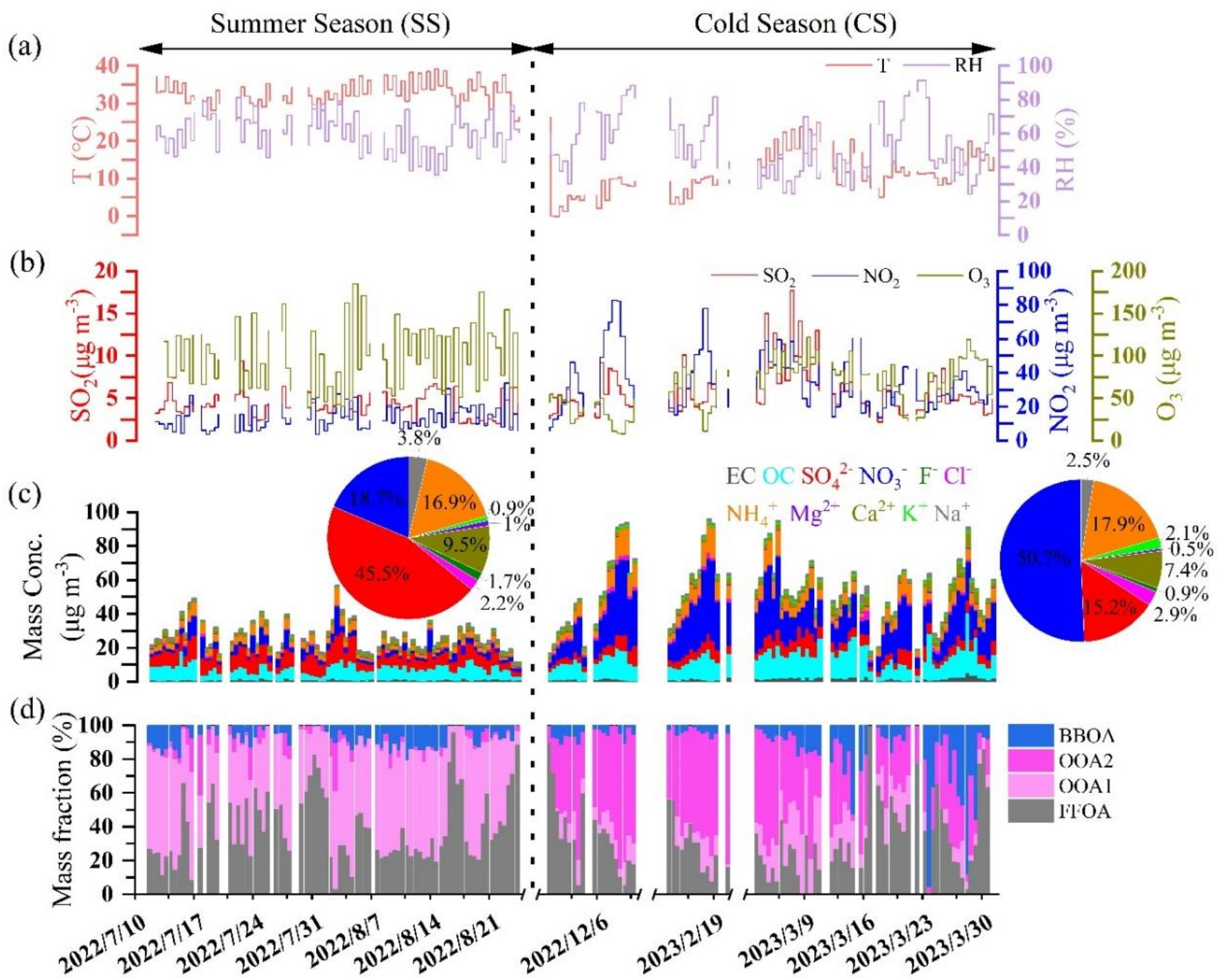

Figure 1. Time series of: (a) air temperature (T) and relative humidity (RH); (b) concentrations of nitrogen dioxide ($NO_2$), sulfur dioxide ($SO_2$) and ozone ($O_3$); (c) concentrations of different inorganic ions, total organic carbon (OC), and elemental carbon (EC) (two inset pies are the average mass contributions of difference ions to the total ions during SS and CS, respectively); and (d) mass percentages of different factors with respect to the total water-soluble OA

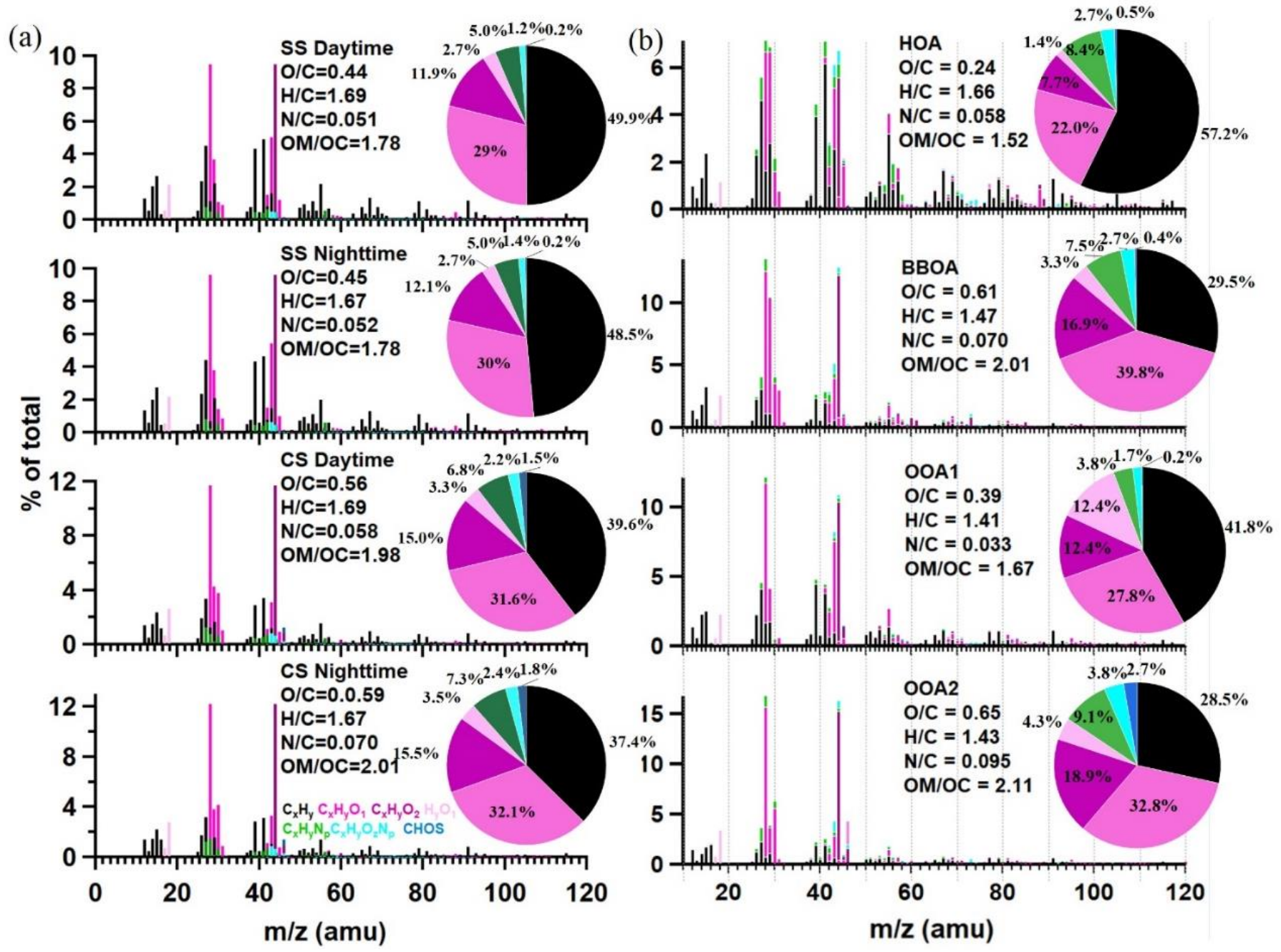

Figure 2. High-resolution mass spectra (HRMS) of (a) the water-soluble OA (WSOA) during different periods, and (b) the four resolved factors (HOA, BBOA, OOA1, OOA2). Ions are classified into and colored by different ion families, and inset pies in both charts show the mass fractional contributions of different ion families to the total HRMS correspondingly.

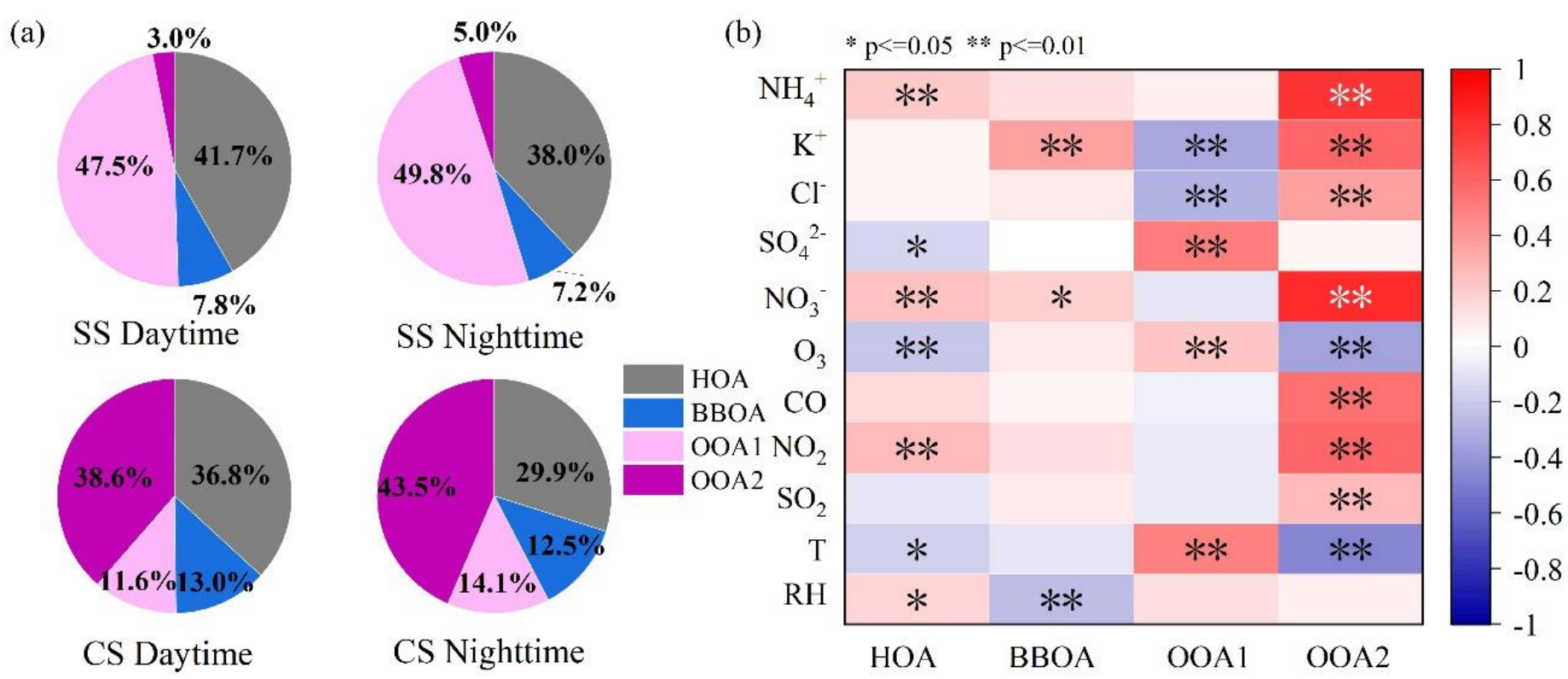

Figure 3. (a) Average mass contributions of the four factor to WSOA during different periods, and (b) cross-correlation coefficients (Pearson's *r*) among the four factors and other aerosol components as well as gaseous species.

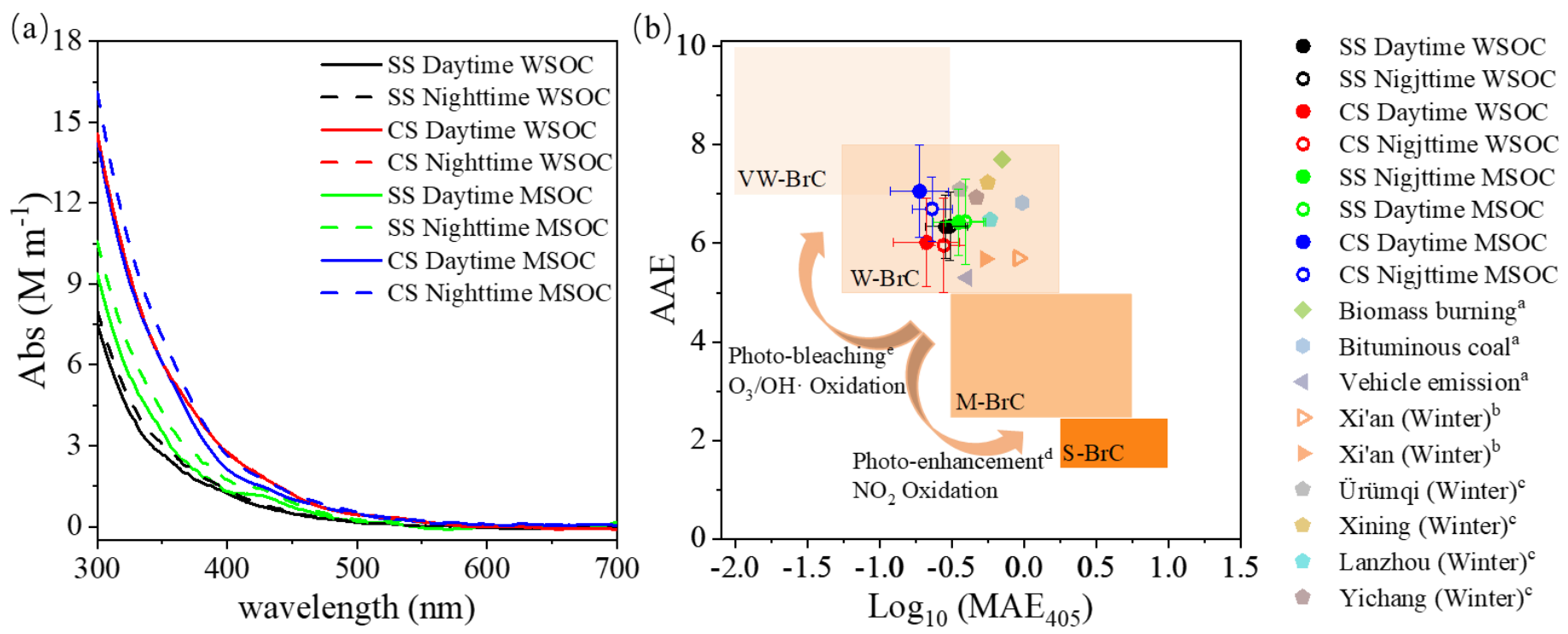

Figure 4. (a) Light absorption coefficients (Abs) of the water-soluble OC (WSOC) and methanol-soluble OC (MSOC) as a function of wavelength, and (b) distribution of the measured data in the $\log_{10}(MAE_{405})$-AAE space (Saleh, 2020)($MAE_{405}$: Mass absorption efficiency at 405 nm; AAE: Absorption Ångström Exponent; The shaded areas indicate very weakly (VW), weakly (W), moderately (M), and strongly (S) absorbing brown carbon (BrC), respectively; Other markers indicate results from [a] Huang et al. (2018), [b] Chen et al. (2018) and [c] Zhong et al. (2023).

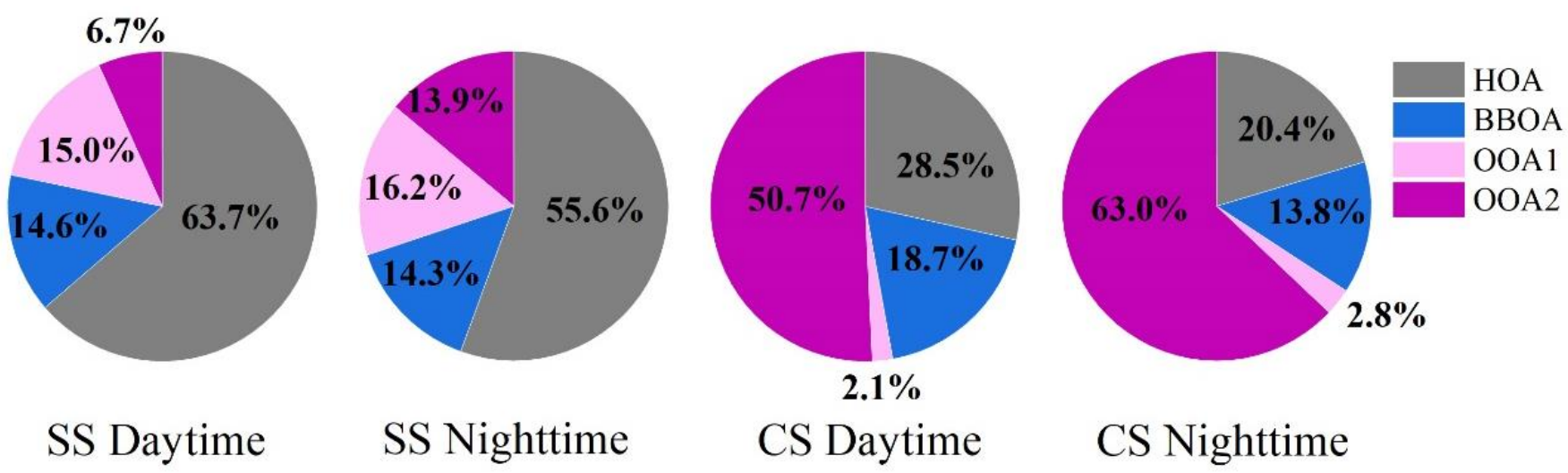

Figure 5. Contributions of the four factors to the total light absorption of WSOA during different periods.

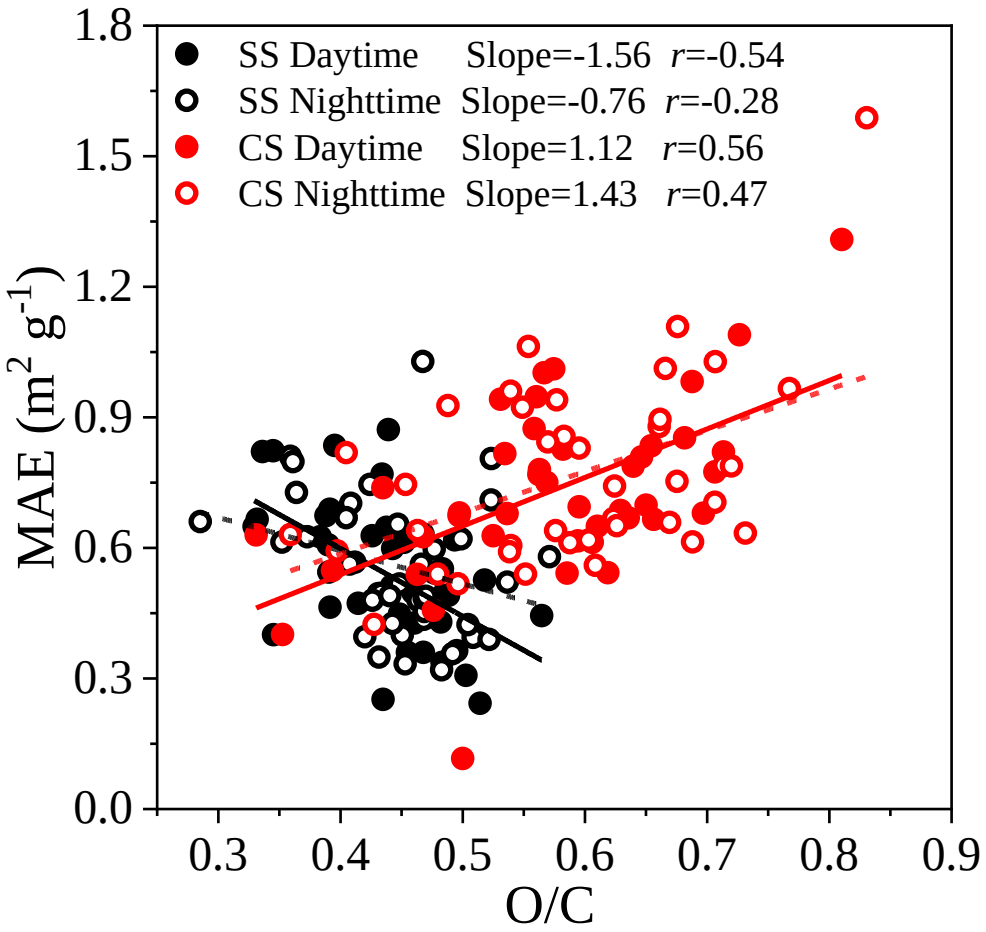

Figure 6. Scatter plot of $MAE_{365}$ (mass absorption efficiency at 365 nm) versus the oxygen-to-carbon (O/C) ratios for the WSOA.

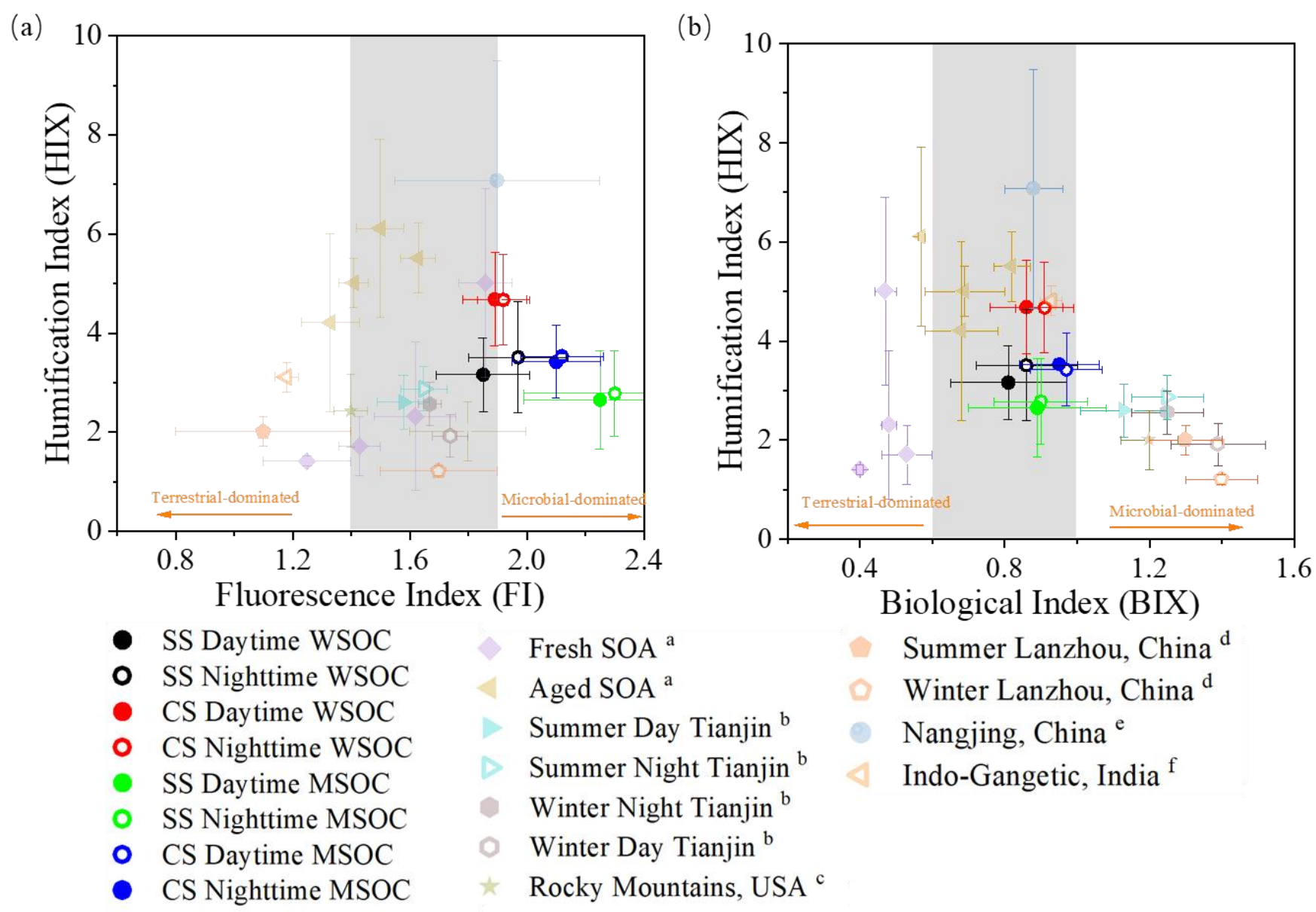

Figure 7. Distribution of the fluorescent indices of measured data in this study and a few other studies ([a] Lee et al. (2013), [b] Deng et al. (2022), [c] Xie et al. (2016), [d] Qin et al. (2018), [e] Xie et al. (2016), [f] Dey et al. (2021)): (a) Humidication index (HIX) versus fluorescenc index (FI), and (b) HIX versus biological index (BIX). The shaded areas marked 0.6 ~ 1 of BIX (Huguet et al., 2009) and 1.6 ~ 1.9 of FI (Mcknight et al., 2001).

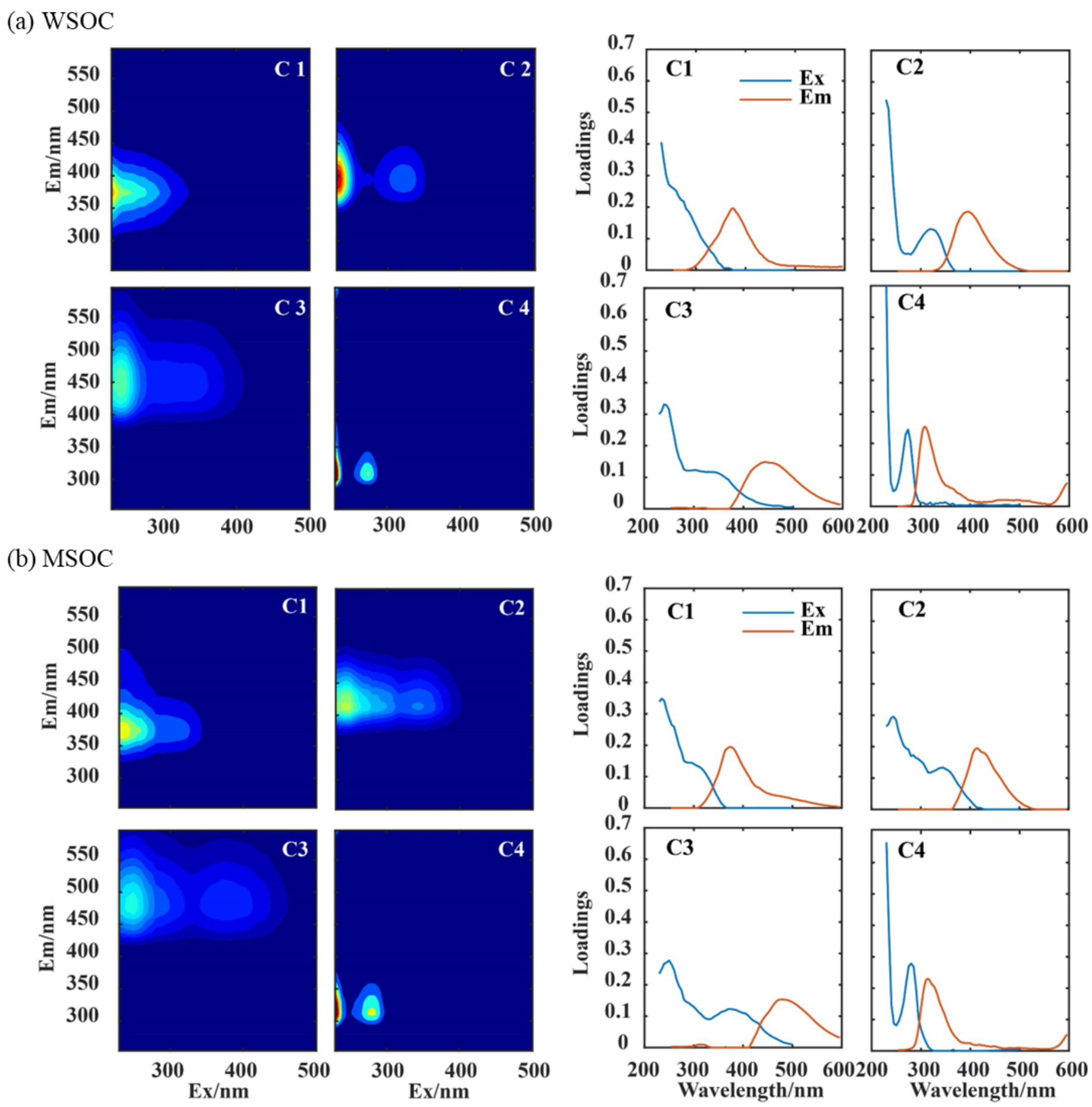

Figure 8. Four fluorescence components (C1 ~ C4) and the corresponding fluorescent intensities of emission (brown) and excitation (blue) against wavelenghth: (a) WSOC, and (b) MSOC.

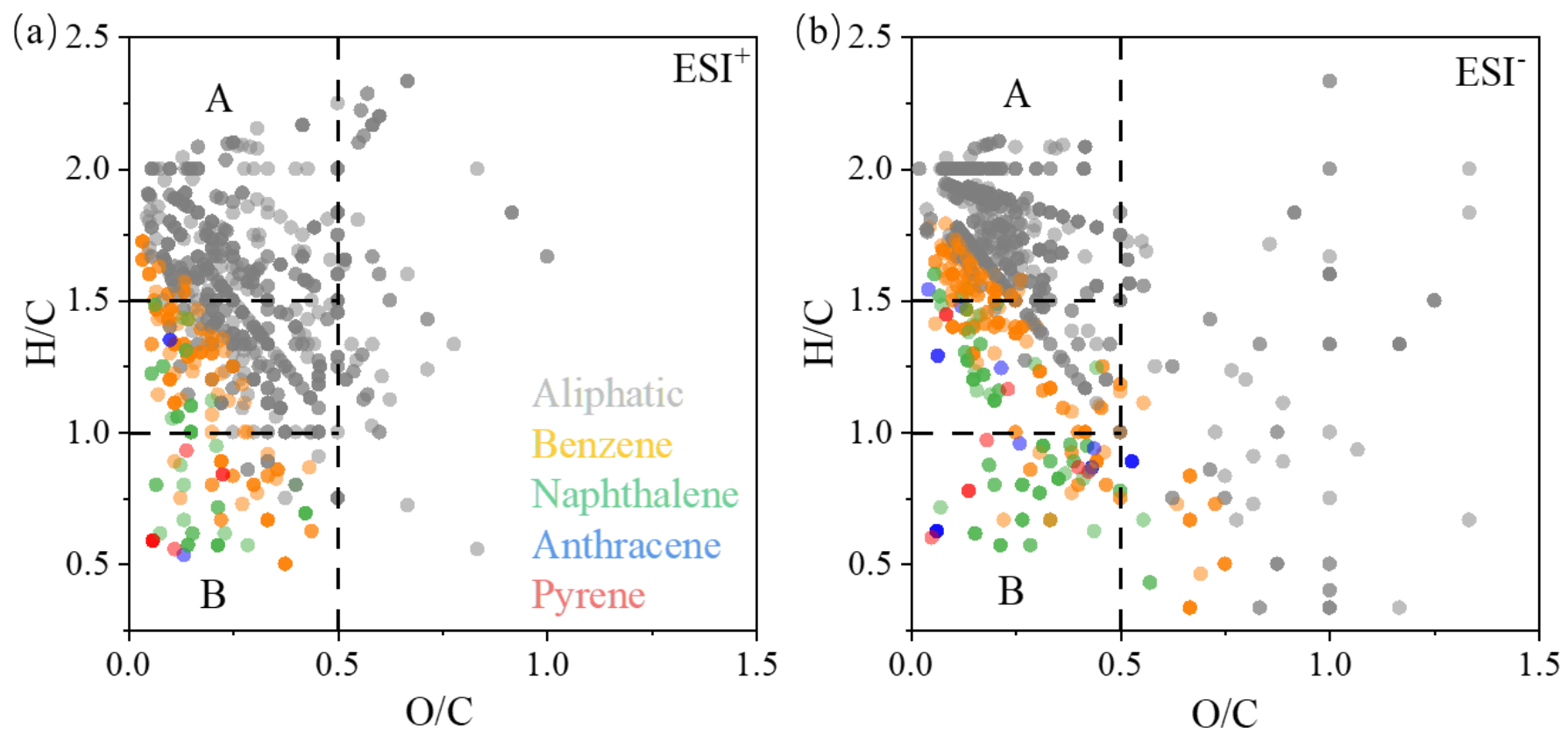

Figure 9. Van Krevelen diagram for CHO compounds detected in (a) $ESI^+$ and (b) $ESI^-$ mode. The markers with different colors represent aliphatic compounds (Xc < 2.50), aromatic benzene ring structures (2.50 ≤ Xc <2.71), naphthalene ring structures (2.71 ≤ Xc <2.80), anthracene ring structures (2.80 ≤ Xc <2.83), and pyrene ring structures (2.83 ≤ Xc <2 .92), respectively (Mao et al., 2022).

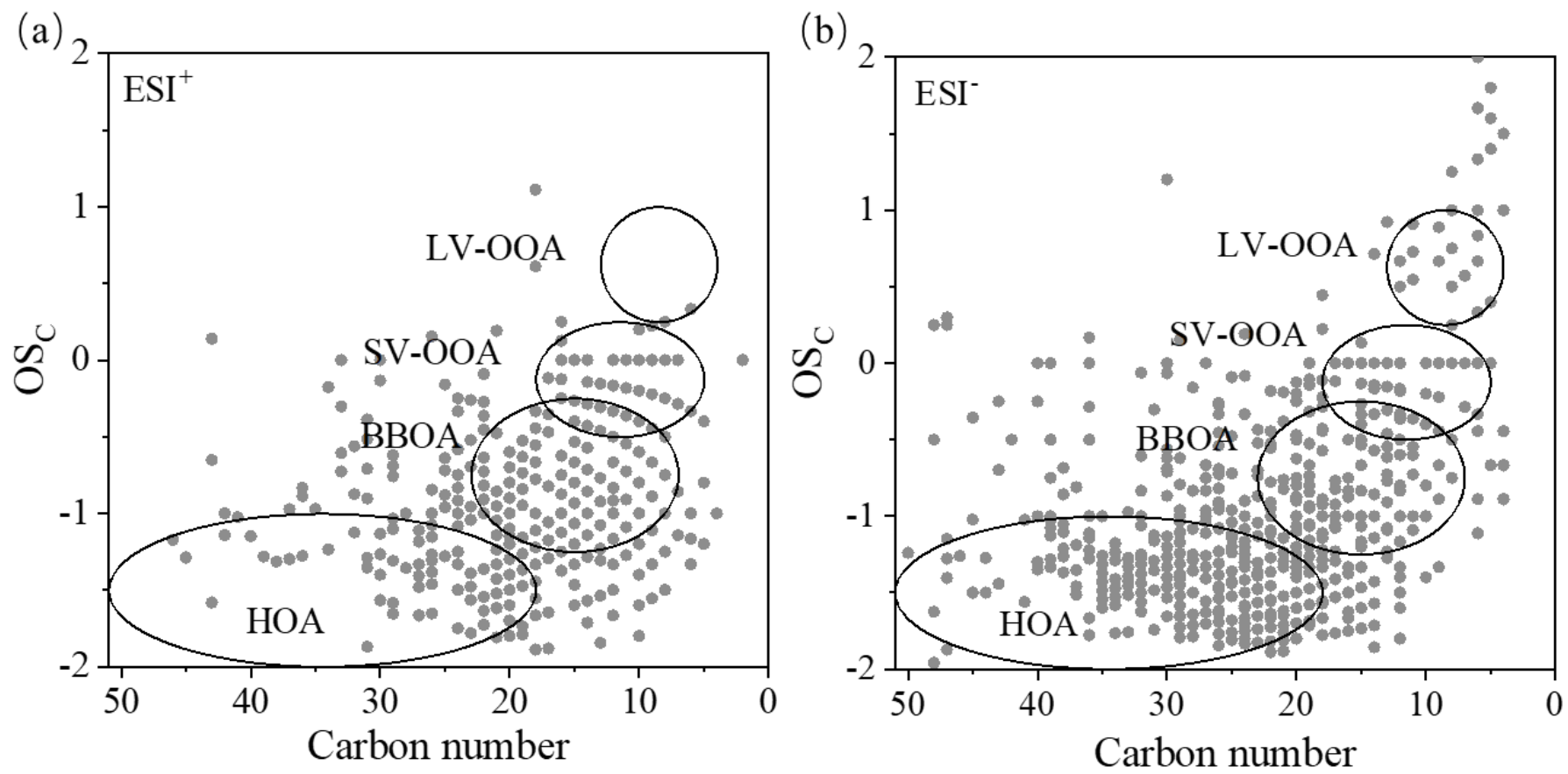

1449

1450 Figure 10. Scatter plots of carbon oxidation state (OSc) versus carbon number for CHO
1451 compounds: (a) $ESI^+$ mode, and (b) $ESI^-$ mode. The circled areas represent those from
1452 fossil fuel combustion hydrocarbon-like OA (HOA), biomass burning OA (BBOA),
1453 semi-volatile oxygenated OA (SV-OOA) and low-volatility oxygenated OA (LV-
1454 OOA)(Kroll et al., 2011).
1455

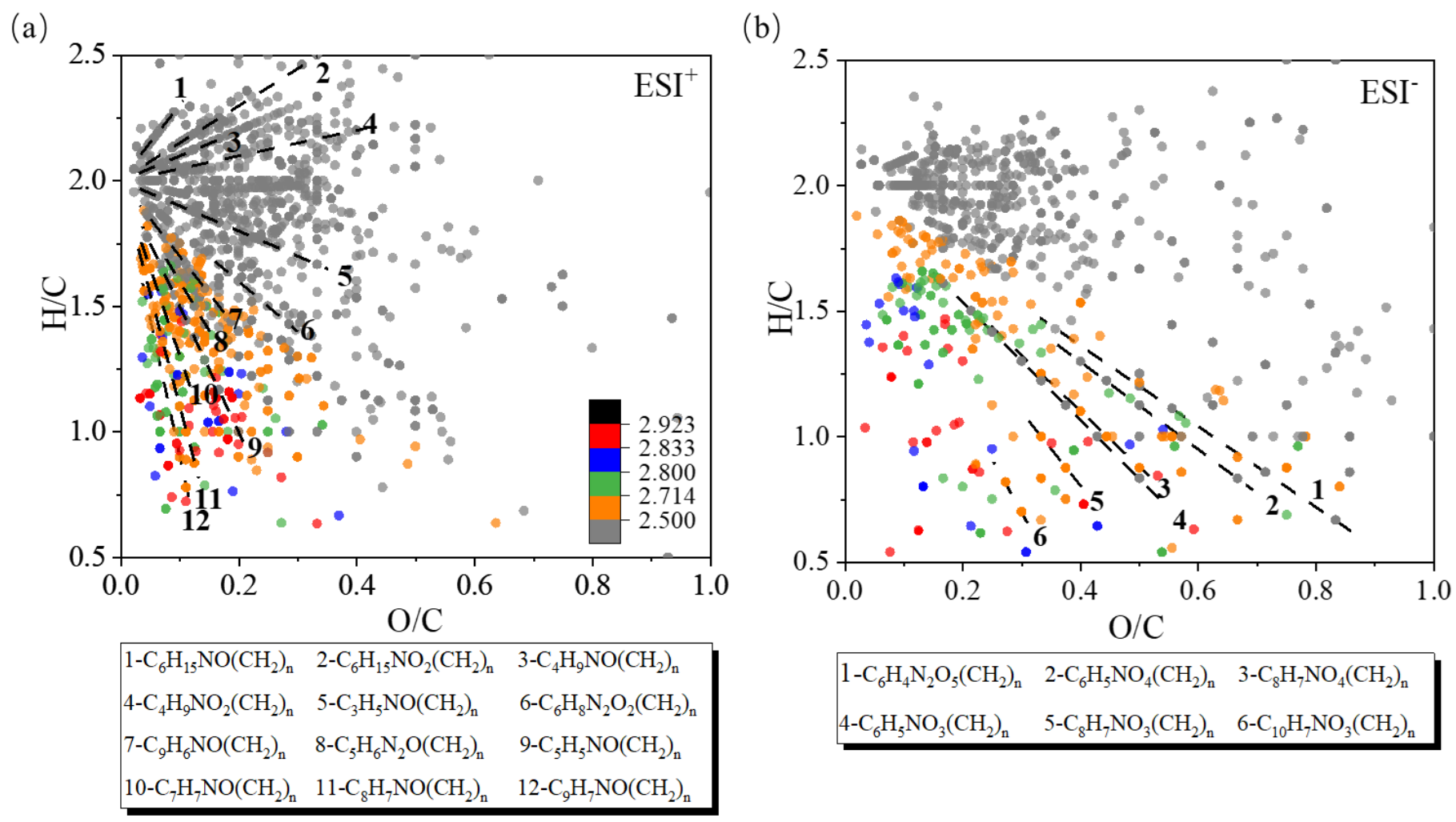

Figure 11. Van Krevelen diagram for CHON compounds detected in both ESI$^+$ and ESI$^-$ mode. The data are also colored by Xc values (See caption of Fig. 9), and the different dash lines represent different series of compounds.

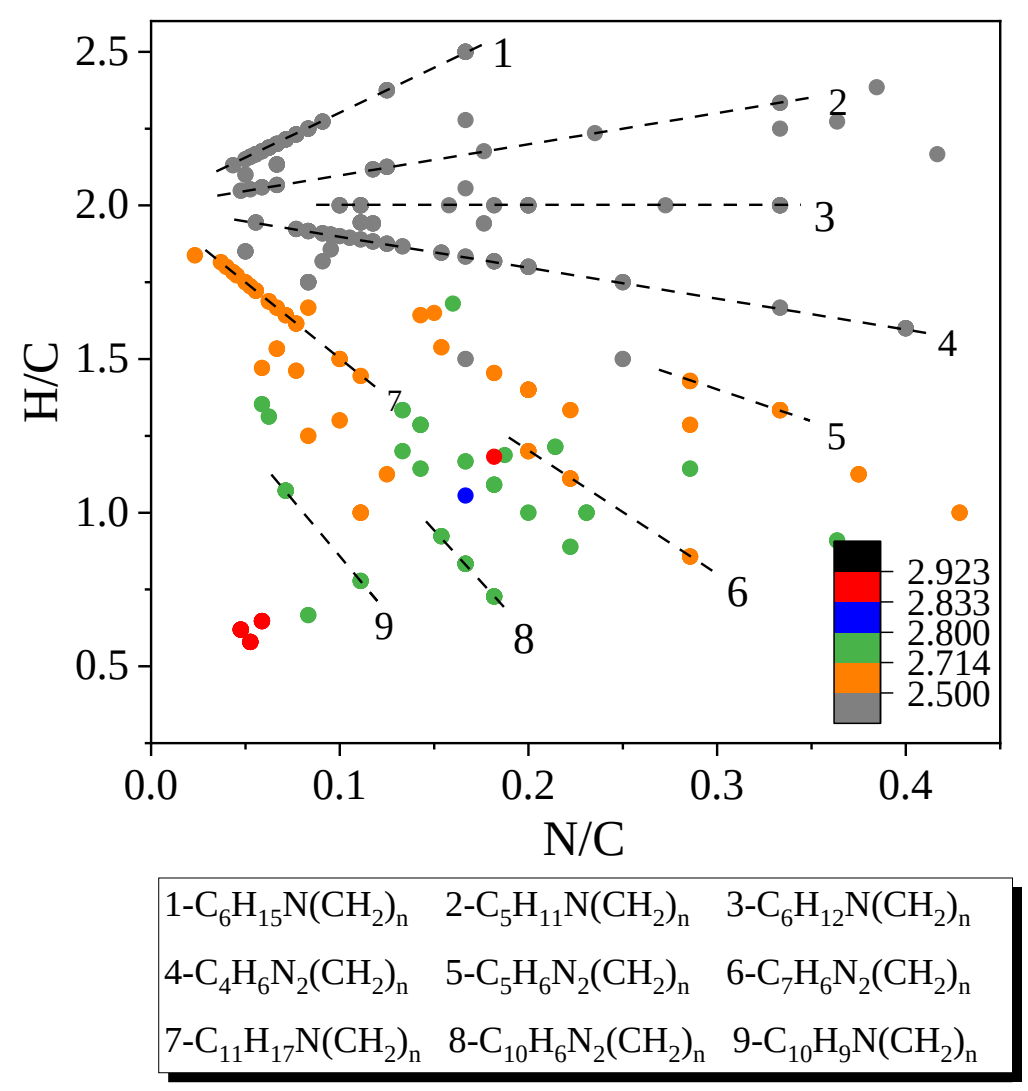

Figure 12. Van Krevelen diagram of the CHN compounds in $ESI^+$ mode. The data are also colored by Xc values (See caption of Fig. 9), and the different dash lines represent different series of compounds.

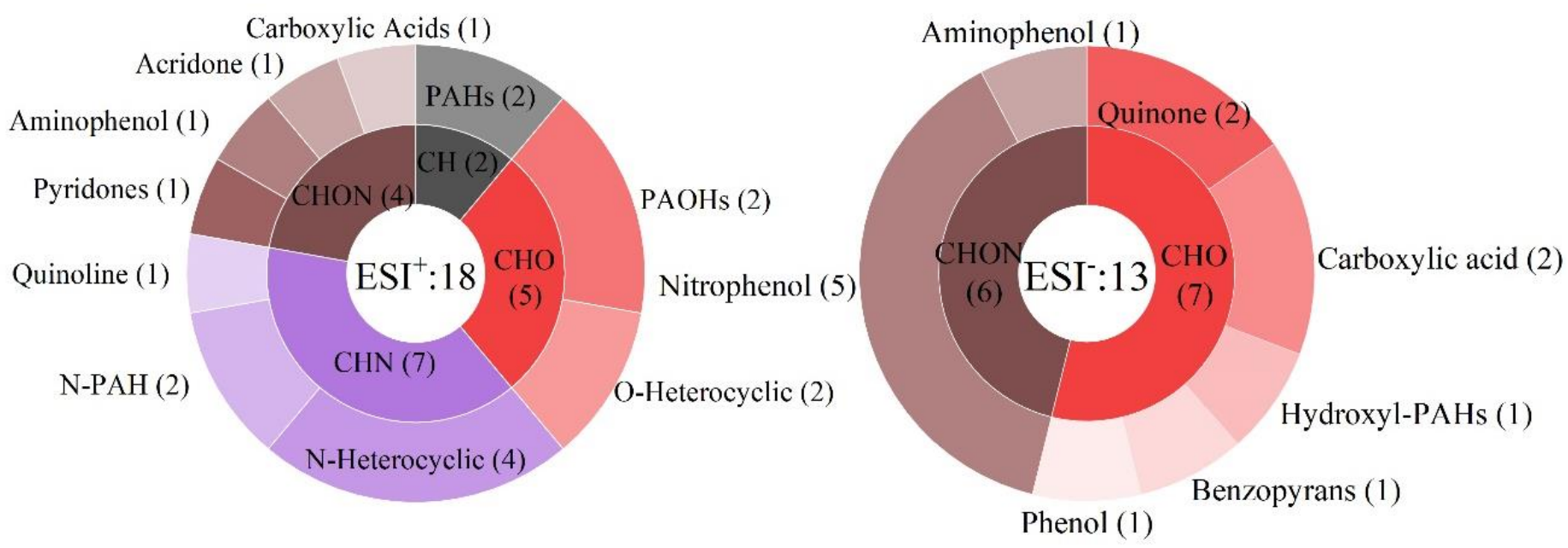

Figure 13. Distributions of the machine learning identified key light absorbing organic compounds (details in Table S2 in the supplement)