# Peer review of "Machine Learning Assisted Chemical Characterization and Optical"

_EGUsphere, 2024_

## Author Comment (AC1)

**To referee #1**

*The manuscript by Huang et al. explores the chemical and optical properties of ambient PM2.5 in an urban area of China. The authors employ a detailed chemical characterization and use a machine learning approach to link optical properties with molecular characterization, providing a valuable reference for future studies. Overall, the manuscript is well-written, and the method is reasonable. I have a few points that could be addressed to strengthen the manuscript:*

**Reply #0:** We thanks a lot for the referee's positive comments on our work. Your specific comments were addressed below.

*The title, "Machine Learning Assisted Chemical Characterization and Optical Properties of Atmospheric Brown Carbon in Nanjing, China," suggests a focus on machine learning. Consider expanding the discussion of this aspect within the manuscript.*

**Reply #1:** Thanks for the suggestion. Yes, the most important novelty of this manuscript is the application of ML method on BrC molecular identification. We have paid attentions to improve the writing to highlight this point throughout the manuscript. Just for one example, we added more details in section 3.4 regarding the ML analysis.

Lines 763-772. "To enhance the robustness of ML analysis, we first incorporated all detected compounds without assigned molecular formulas of both positive and negative modes (4953 formulas, ESI$^+$: 2863, ESI$^-$: 2090) in Section 3.3 into the ML model. After ML analysis, a total of 1477 molecules (ESI$^+$: 795; ESI$^-$: 682) were found to have positive values for both IncNPu_val and IncMSE_val. Among them, 1051 molecules (ESI$^+$: 420; ESI$^-$: 450) met the criteria and were assigned corresponding molecular formulas; and furthermore, a total of 149 compounds with 0.5<DBE/C<0.9 (ESI$^+$: 52; ESI$^-$: 97) were chosen. By comparing with the database, we finally proposed 31 compounds (ESI$^+$: 18, ESI$^-$:13) as the key BrC species; details regarding their molecular formulas and proposed structures, etc., are summarized in Table S2."

*Since the authors also examine the fluorescent properties of organic aerosols (OA), it would be beneficial to include a brief introduction to this topic in addition to the discussion on light absorption.*

**Reply #2:** As suggested, we have added a few sentences regarding the fluorescence properties of aerosols in the revised manuscript,

Lines 97-103, "On the other hand, the fluorescent properties of OA can be determined by the excitation-emission matrix (EEM) fluorescence spectroscopy (Murphy et al., 2013; Stubbins et al., 2014). By performing the parallel factor analysis (PARAFAC) on EEM data, the key fluorophores can be identified (Xie et al., 2020; Chen et al., 2020a; Chen et al., 2021). These fluorophores are also linked with different sources such as biomass burning, coal combustion, and vehicle emissions (Tang et al., 2020)."

*Add more comparisons with similar offline PMF-OA analyses to strengthen and justify the robustness of your PMF results on WSOA.*

**Reply #3: As suggested,** we have added more discussions of other offline PMF results in Section 3.1.2, by adding a few new studies using the offline AMS analysis techniques.

Lines 380-382, "This is consistent with the behaviors of HOA from a number of previous offline AMS results ( Daellenbach et al., 2017; Liu et al., 2021; Qiu et al., 2019; Ye et al., 2017)."

Lines 390-393, "Note the BBOA here had a relatively higher O/C ratio of 0.61 than those identified in previous offline AMS measurements, such as Yangzhou (0.45) (Ge et al., 2017a), Beijing, China (0.59) (Qiu et al., 2019), and Marseille, France (0.54) (Bozzetti et al., 2017), suggesting the presence of partially aged BBOA species in this factor. "

Lines 417-423, "It should be noted that, BBOA can be more important than HOA, such as in Bejing during autumn polluted period (38.3% of WSOA) (Hu et al., 2020), in Delhi, India (31-34% of WSOA) (Bhowmik et al., 2024), and in urban and rural Catalonia, Spain (up to 26% of WSOA) (Veld et al., 2023). In general, the dominance of SOA in WSOA observed here was consistent with most offline AMS studies aforementioned, except that POA was found to dominate WSOA in Delhi, India (Bhowmik et al., 2024)."

*Clarify the reasoning for using MAE365 initially and MAE405 in Figure 4b. This discrepancy should be explained.*

**Reply #4:** Yes, Figure 4b uses MAE405 as this is the value recommended by the references as the indicator to classify different types of BrC (VW-BrC, W-BrC, M-BrC and S-BrC). On the other hand, the MAE365 was recommended to a better value to

represent the light absorption properties of BrC without significant interferences from other species. In, we therefore also calculated the MAE405 here. MAE405 here was only used in Figure 4b for the convenience of direct comparison with other results, and does not affect other discussions in the manuscript.

*The turning point in Figure 6 is unclear. Please provide additional explanations to enhance understanding.*

**Reply #5:** As suggested, we have added more discussions regarding this point.

Lines 563-569, "Many studies have shown that aqueous reactions are a source of BrC (e.g., ), but evidence also shows that further aqueous ageing of BrC can lead to photo-bleaching (e.g., ), therefore a certain O/C turning point can exist if the OA evolution was governed by a certain process. The real atmospheric processes are however complicared, such O/C points might be less clear. Nevertheless, similar points were indeed observed previously (Zhong et al., 2023; Jiang et al., 2022)."

*Discuss any structural similarities among the key BrC species identified using the machine learning RF approach, as this would add value to the findings.*

**Reply #6:** Thanks for the suggestion, this is indeed a good point. We have added a discussion on the structural similarities among the identified BrC species in section 3.4 Lines 813-824, "For the BrC species identified here, most species contain at least one benzene ring (such as PAHs and NACs) except two compounds (2-hydroxypyridine and urocanate) with other aromatic ring structures. This is reasonable as organic compounds with benzene or other aromatic rings (with conjugated double bonds) are known to be strong light-absorbing. The -OH and -COOH groups on the benzene ring can enhance ultraviolet light absorption at near-UV wavelengths (Jacobson, 1999); The -$NO_2$ group can further increase light absorbance at longer wavelengths (Wang et al., 2020). Our identified BrC list does include compounds with such functional groups. Note some nitrogen-containing heterocyclic compounds, are usually secondary products of aqueous reactions between carbonyl compounds (such as glyoxal and methylglyoxal) and amines or ammonium (Powelson et al., 2014); thus identification of them here as key BrC species is also a supporting evidence of the occurrence of aqueous reactions."

*Increase the font sizes in the figures to improve readability.*

**Reply #7:** As suggested, we have increased the font sizes in some Figures.

**To referee #2**

*General comments:*

*This study provides a detailed investigation into the chemical and optical properties of brown carbon (BrC) in ambient PM2.5 samples collected in Nanjing, China. It sheds new light on the chemical, molecular, and optical characteristics of BrC. The paper is well-written, logically organized, and includes appropriate figures and tables. However, there are major errors or gaps that need to be addressed. With significant revisions, additional data, and/or other changes, it has the potential to make a meaningful contribution.*

*Major:*

*1/I am unclear about the innovation and contributions of this study after reading the abstract and introduction. I suggest that the authors clearly articulate the limitations of existing research and the potential contributions of this study in the abstract and background sections. For example, in the third paragraph (Lines 91-92), the authors state, 'the aforementioned identified species only account for a limited fraction of BrC total light absorption.' Can we then infer that this study has identified additional BrC species contributing to light absorption?*

**Reply #8:** Thank you for your suggestion. We agree that a clear statement about the limitations of existing research and novelty of this study in the abstract and introduction sections can better convey the new findings to the readers and the science community. Per your suggestion, we have added such descriptions in the abstract and introduction, as shown below.

Lines 20-26, in the abstract, "yet current knowledge of chemical composition of BrC is largely limited to a few certain classes of compounds; the chemical and optical properties, and particularly linkage between the two remain poorly understood. To address this, a comprehensive investigation was conducted on the particulate matter (PM$_{2.5}$) samples collected in Nanjing, China during 2022 ~ 2023 with a focus on identification of the key BrC molecules."

Lines 127-131, in the last paragraph of introduction, "Particularly, for the first time, we applied the ML RF algorithm to connect the light-absorbing characteristics with the determined organic molecules, aiming to identify more unknown key BrC molecules.

Our findings regarding the BrC properties, especially the BrC molecules proposed here can be a good reference, and the ML application can be an example of practice for future studies as well."

At last, the answer to your question is "Yes", we indeed discovered new BrC species that are not reported previously to the best of our knowledge. Therefore, our practice with ML to find BrC species here, can also be an example and likely calls for more similar research in the future, therefore enhance our scientific understanding on BrC. This point is now also mentioned in the conclusions part,

Lines 872-875, "However, our practice with ML approach here serves as only a case study and a valuable attempt, more studies likely with more advanced ML algorithms are needed to identify more BrC species, and to achieve a quantitative closure between the BrC molecules and its total light absorption."

*2/My second major concern is with the results. I noticed that the mass concentration and absorption of MSOC are higher than those of WSOC. Why didn't the authors analyze the sources and characteristics of MSOC in Section 3.1.2?*

**Reply #9:** Thanks for the comment, we understand your concern here.

First, indeed, it is better if we are able to analyze the sources of MSOC in Section 3.1.2. However, due to the technical limitation, it is not feasible for now. There is no effective way to quickly remove the methanol solvent, the signals we intend to obtain for the OA dissolved in methanol would be very significantly interfered by the methanol itself and the organic impurities in methanol, making the results of MSOA unreliable. We have tried to do so by using active carbon to remove methanol (similar to that we used silica gel to remove water), but the effects are unrealistic. Therefore, we can now only analyze the WSOC rather than MSOC. In fact, this is the reason that why most AMS offline analyses available in the literature applied only on water-soluble OA. Secondly, WSOC and MSOC correlated very well as shown in Fig. S5, suggesting a large overlap between WSOC and MSOC. At last the ML application is for light absorption of MSOC and molecular characterization of MSOC, it is consistent. Therefore, source analysis results of WSOC rather than MSOC, do not affect major findings of this work. The technical development regarding the offline AMS analysis on MSOC is the subject of our future work. This is indeed one of the limitation of this study, and it has been mentioned in the conclusion section (in **Reply #12** too)

The information above is now made clear in the revised manuscript,

Lines 177-183, "It should be noted that, direct AMS analysis on the methanol-soluble OA (MSOA) is currently unfeasible, even though it might be more important than WSOA in both concentration and light absorption. Likewise almost all offline AMS analysis methods (O'Brien et al., 2019; Vasilakopoulou et al., 2023), this is due to that the methanol solvent and its associated organic impurities cannot be effectively removed, making the obtained MSOA signals unidentifiable."
Lines 304-305, "Note both light absorption and organic molecules are for MSOA."
Lines 865-868, "One limitation is that no source apportionment was conducted on MSOA due to technical difficulties, therefore direct link between emission sources and optical properties are incomplete; further development of proper method should be the subject of our future work."

*3/In Line 484, I am unsure how the authors reconstructed Abs365, WSOC. I suggest that the authors explain the reconstruction of Abs365, WSOC in the Methods section or in the supplementary materials.*

**Reply #10: Per your suggestion, we have added it in Section 2.3.1**

Lines 260-266, "Additionally, a multi-linear regression (MLR) method was used to estimate the contributions of different WSOA factors to the light absorption of total WSOA, as shown in the following equation :

$$Abs_{365,WSOC} = a \times HOA + b \times BBOA + c \times OOA1 + d \times OOA2 \qquad (4)$$

Here, HOA, BBOA, OOA1, and OOA2 represent time series of the WSOA factors. *a*, *b*, *c*, and *d* are the fitting parameters, which are the mass absorption efficiency (MAE) values of corresponding factor."

*4/In Section 3.2.4, the authors mention that the fluorescent properties of WSOC and MSOC are governed by HULIS during CS, while they are governed by a protein-like component during SS. What does this mean? What is the connection between HULIS and the protein-like component in relation to the sources? I suggest discussing this in the Results section.*

**Reply #11: Thanks for the suggestion. Overall, the protein-like component (C4) mainly come from biological activity, therefore its emission can be enhanced during SS,

leading to a higher contribution. HULIS components have much complicated sources, C1 is linked with combustion activities, C2 might be a mix of both primary and secondary sources, while C3 is strongly linked with highly oxidized organics therefore from aqueous/heterogeneous reactions in this study. These information are now discussed in the revised manuscript.

Lines 615-616, "C1 exhibited a peak at Ex = 230 nm and Em = 374 nm, identified as less oxidized HULIS typically associated with combustion sources."

Lines 623-624, "….likely a mix of primary and secondary sources. C2 contribution was thus comparable during different seasons (37.5 % vs. 38.6 %)"

Lines 626-627,631-633 "C3 component, with a peak at Ex = 240 nm and Em = 446 nm, was considered as a highly oxidized HULIS component, relevant with secondary processes…As discussed earlier, aqueous/heterogenous reactions contributed to WSOA, especailly during CS; correspondingly, C3 contribution was indeed much higher during CS than during SS (24.1 % vs. 14.7 %)."

Lines 647-651, "The increased contributions of the protein-like fluorophore (C4) during SS in both WSOC and MSOC were likely due to its major origin of biological activities, which can be enhanced by the relatively high temperatures during SS (Fan et al., 2020); this is likely the reason that C4 contributions during daytime were higher than those during nightime in both SS and CS."

*5/I would suggest adding a discussion on the limitations of this study and the uncertainties associated with the methods or equipment in the conclusion section.*

**Reply #12:** Agreed. It is valubale to the readers to lay out the limitations of our study. We have added a discussion in the conclusions section.

Lines 865-875, "Of course, our study has some limitations. One limitation is that no source apportionment was conducted on MSOA due to technical difficulties, therefore a full closure between emission sources and optical properties of total OA is incomplete; development of proper method should be the subject of our future work. Moreover, our findings here expand the understanding on chemical (both bulk and molecular level) and optical properties (both light absorption and fluorescence) of BrC, and are valuable to evaluate its impact on air quality and radiation balance; the identified list of key BrC molecules can be a useful reference for future studies.

However, our practice with ML approach here serves as only a case study and a valuable attempt, more studies likely with more advanced ML algorithms are needed to identify more BrC species, and to achieve a quantitative closure between the BrC molecules and its total light absorption."

*Minor:*
*The text and figures require a thorough review for general errors. e.g., the last pannel in Fig 2a, 0.0.59 -> 0.59?*

**Reply #13:** We are very sorry for the typos. A thorough check on the tables and figures was performed to avoid any errors.

*I would suggest adding a sentence to transition after 'Here, a comprehensive analysis was conducted on the particulate matter (PM2.5) samples collected in Nanjing, China, during 2022–2023, with a particular focus on the identification of key BrC molecules' (Lines 21-23). For example, 'Several important clues related to BrC were found.'*

**Reply #14:** Revised as suggested.

*I would suggest the authors cite the newest studies to illustrate the importance of BrC in the first pargraph of introduction.*
*Here i recommend some,*
*Chakrabarty, R. K., N. J. Shetty, A. S. Thind, et al., 2023: Shortwave absorption by wildfire smoke dominated by dark brown carbon. Nature Geoscience, 2023, 16(8): 683-688.*
*Brown, H., H. Wang, M. Flanner, et al., 2022: Brown carbon fuel and emission source attributions to global snow darkening effect. Journal of Advances in Modeling Earth Systems, 14, e2021MS002768.*
*DeLessio, M. A., K. Tsigaridis, S. E. Bauer, et al., 2023: Modeling atmospheric brown carbon in the GISS ModelE Earth system model. EGUsphere, 1-50.*
*Xu, L., Lin, G., Liu, X., Wu, C., Wu, Y., & Lou, S. (2024). Constraining light absorption of brown carbon in China and implications for aerosol direct radiative effect. Geophysical Research Letters, 51, e2024GL109861. https://doi.org/10.1029/2024GL109861*

**Reply #15:** A few latest studies have now been added in the introduction, including the useful literature you recommended (they are now all cited in proper positions). Please check the revised manuscript.

---

## Author Response (AR2)

**To referee #2**

*1/ In Line 62, I noticed that the reference Wang et al., 2021 appears to be misplaced in the text. It would be more appropriate to include it within the next citation bracket.*
**Reply #1:** Thanks. It is now revised as suggested.

*2/ I recommend incorporating more recent literature to support the discussion, particularly in line 70, where a reference from 2006 is cited.*
**Reply #2:** Thanks. We have now cited a literature from 2024.
Liu, S. J., Wang, Y. Q., Zhang, S., Chen, Y. B., Wu, C., Zhang, G. Q., and Wang, G. H.: The synergistic effect of NOx and SO2 on the formation and light absorption of secondary organic aerosols from o-xylene photooxidation, Atmos. Res., 304, https://doi.org/10.1016/j.atmosres.2024.107387, 2024.

*3/ In Line 304-305, "Note both light absorption and organic molecules are for MSOA.",  Should this term be "WSOA" instead of "MSOA"? In your conclusion part, i saw one limitation of this paper is that no source apportionment was conducted on MSOA due to technical difficulties, please check for this.*
**Reply #3:** Sorry that we did not state it clearly, it is indeed "MSOA" not "WSOA". Here we want to clarify that since the molecular characterization of OA was conducted on methanol-soluble OA (MSOA) not WSOA, therefore the light absorption data was also for MSOA. The source apportionment was conducted on WSOA not MSOA due to technical difficulties, but the source apportionment results were not used for ML analysis therefore do not affect the ML results. We change the sentence to "Note since molecular characterization was conducted on MSOA (not WSOA), therefore light absorption of MSOA were used here for consistency."

*4/ I would suggest expanding the discussion of the machine learning results in the conclusion section, as this represents a significant innovation of your work. Specifically, it would be valuable to highlight which findings were uniquely discovered or validated through the machine learning approach. Providing more detailed insights into these aspects would strengthen the impact of your study and better emphasize its contributions to the field.*
**Reply #4:** Thanks for the suggestion. We have now expanded our discussion on the ML results, it now reads, "Nevertheless, the ML approach has demonstrated its great potential in identifying new key BrC species (such as 4-methylcoumarin, urocanate, etc here) as well as reaffirming the important role of known key BrC molecules (such as the nitrogen-containing aromatic molecules; such ML-identified list can be a useful reference for future studies. Of course, since there are multiple types of ML algorithms (such as supervised, unsupervised, semisupervised and reinforcement learning) with differing performances as well as advantages/pitfalls used in environmental research (Zhu et al., 2023), the rigorous, accurate, robust and also practicable ML analysis requires more research efforts and should be an active and important topic in the future. The ML approach can be a powerful and promising tool to achieve a quantitative closure between the BrC molecules and its total light absorption. "